# Universality class of the glassy random laser

Jacopo Niedda,[1, 2] Giacomo Gradenigo,[3, 4, 2] Luca Leuzzi,[2, 1, *] and Giorgio Parisi[1, 2, 5, 6]

[1]*Dipartimento di Fisica, Università di Roma "Sapienza", Piazzale A. Moro 2, I-00185, Roma, Italy*
[2]*NANOTEC CNR, Soft and Living Matter Lab, Roma, Piazzale A. Moro 2, I-00185, Roma, Italy*
[3]*Gran Sasso Science Institute, Viale F. Crispi 7, 67100 L'Aquila, Italy*
[4]*INFN-Laboratori Nazionali del Gran Sasso, Via G. Acitelli 22, 67100 Assergi (AQ), Italy*
[5]*INFN, Sezione di Roma-1, P.le A. Moro 5, 00185, Rome, Italy*
[6]*Accademia Nazionale dei Lincei, Palazzo Corsini - Via della Lungara, 10, I-00165, Roma, Italy*

By means of enhanced Monte Carlo numerical simulations parallelized on GPUs we study the critical properties of the spin-glass-like model for the mode-locked glassy random laser, a 4-spin model with complex spins with a global spherical constraint and quenched random interactions. Implementing two different boundary conditions for the mode frequencies we identify the critical points and the critical indices of the random lasing phase transition with finite size scaling techniques. The outcome of the scaling analysis is that the mode-locked random laser universality class is compatible with a mean-field one, though different from the mean-field class of the Random Energy Model and of the glassy random laser in the narrow band approximation, that is, the fully connected version of the present model. The low temperature (high pumping) phase is finally characterized by means of the overlap distribution and evidence for the onset of replica symmetry breaking in the lasing regime is provided.

## I. INTRODUCTION

When light propagates through a random medium, scattering reduces information about whatever lies across the medium and the electromagnetic field, composed by many interfering wave modes, provides a complicated emission pattern as light undergoes multiple scattering. If enough power is pumped into the medium multiple scattering may support the population inversion of atoms and molecules above some optical gap, yielding a random laser [1–14]. Random lasers are made of an optically active medium and randomly placed scatterers (sometimes both in one [2]). The first provides the gain, the latter provides the high refraction index and the feedback mechanism needed to lead to amplification by stimulated emission. As opposed to ordered standard multimode lasers, random lasers do not require complicated construction and rigid optical alignment, have a low cost, undirectional emissions, high operational flexibility and give rise to a number of promising applications in the field of speckle-free imaging [15, 16], granular matter [9, 10], remote sensing [13, 17, 18], medical diagnostics and biomedical imaging [13, 19–22], optical amplification and optoelectronic devices [13, 23, 24].

Random lasers may show multiple sub-nanometer spectral peaks above a pump threshold [2], as well as smoother, though always disordered, emission spectra. Depending on the material, and its optical and scattering properties, random spectral fluctuations between different pumping shots (i. e., different realizations of the same random laser) may or may not vary significantly. A wide variety of spectral features is reported [10, 12, 25–27], depending on material compounds and experimental setups. Random lasers can be built in very different ways,

can be both solid or liquid, can be 2D or 3D, the optically active material can be confined or spread all over the volume. Moreover, random lasers are, usually, open systems where light can propagate in any direction rather than oscillating between well specific boundaries (mirrors) as in standard lasers and the emission acquisition can only be directional rather than on the whole solid angle. Finally, also the scattering strength and the pumping conditions may affect the emission.

In the last years experiments on a certain class of random lasers provided evidence of particularly non-trivial correlations between the shot-to-shot fluctuations of the emission spectra. We will refer to those as *glassy* random lasers [13, 28–32]. These special correlations are predicted by a theory based on statistical mechanics of complex disordered systems [33–35]. Indeed, it has been shown that these fluctuations are compatible with an organization of mode configurations in clusters of states, similar to the one occurring for complex disordered systems displaying multiequilibria, as the spin glasses. Such a correspondence has been analytically explained proving the equivalence between the distribution of the Intensity Fluctuation Overlaps (IFO) and the distribution of the overlap between states, the so-called Parisi overlap, the order parameter of the glass transition [36]. Though the analytical proof assumes narrow-band spectra, such that all modes - within their line widths - can be considered at the same frequency [33, 37, 38], numerical simulations have provided evidence that the onset of nontrivial distributions of IFO and Parisi overlap distributions occur at the same (critical) temperature also in realistic models for random multimode lasers [39]. In these models, the four-waves non-linear mixing between electromagnetic field modes is controlled by a deterministic selection rule depending on modes frequencies, termed *mode-locking*. In mode-locked lasers interactions are possible only for the quadruplets of modes whose frequencies

* luca.leuzzi@cnr.it

$\omega_k$ satisfy the condition

$$|\omega_{k_1} - \omega_{k_2} + \omega_{k_3} - \omega_{k_4}| < \gamma, \qquad (1)$$

with $\gamma$ being the typical line-width of the modes. We will refer to Eq. (1) as Frequency Matching Condition (FMC). In standard mode-locked lasers such selection rule is implemented by ad hoc nonlinear devices (e.g., saturable absorbers for passive mode-locking [40]) that are not there in random lasers. As hypothesized in [41] and recently experimentally demonstrated in [12], though, in random lasers mode-locking occurs as a self-starting phenomenon. We call an interaction network built on the mode-locking selection rule in Eq. (1) a Mode-Locked (ML) graph.

From the point of view of statistical mechanics of complex disordered systems, random lasers represent, so far, the only physical system where the relevant degrees of freedom, namely the complex amplitudes of the light modes, naturally form a dense interaction network of the kind for which replica symmetry breaking mean-field theory [42] is proved to work, as in high dimension spin-glasses or structural glasses made of hard spheres [43]. It is not by chance that random lasers are, so far, the only complex disordered system providing experimental evidence of a continuous replica symmetry-breaking pattern [13, 28–32]. Actually, mean-field theory for an infinite number of replica symmetry breakings has rigorously been derived [44, 45] only for fully connected systems, including the random laser model in the narrow-band approximation [33, 38]. Using the cavity method it is, then, possible to compute a replica symmetry breaking (RSB) phase also in systems with sparse interactions[46] (the Viana-Bray model, for instance [47–49]). Still, the correct mean-field theory which describes ML random lasers has yet to be found, due to some peculiarities of the interaction between light modes that will be detailed in the following.

In this work we resort to Monte Carlo numerical simulations of the dynamics of a leading model for multimode random lasers, the Mode-Locked (ML) 4-phasor model [33, 34, 38, 50].

Even if the phenomenology of the model is quite rich already in the narrow bandwidth approximation, going beyond the fully-connected case is necessary to achieve a realistic description of random lasers in the spin-glass theoretical framework. If $N$ is the number of modes, the FMC leads to $O(N)$ dilution in the interaction graph: the total number of interactions, which is of order $O(N^4)$ in the complete graph, is, thus, reduced to $O(N^3)$ in the diluted graph [51].

Therefore, as far as the the interaction graph is concerned, the ML 4-phasor model places itself in an intermediate position between the complete and the sparse graph, the latter being the case where the number of couplings per variable does not scale with $N$ in the thermodynamic limit. The analytical solution of a spin-glass model in such an intermediate regime of dilution is a very hard problem to address, since standard mean-field techniques such as RSB theory, [52, 53], do not straightforwardly apply and the cavity method for sparse [48] or diluted dense networks [54] does not allow to devise close equations for global order parameters and provide a fully explicit solution. Eventually, to the best of our knowledge, no spin-glass model has been solved exactly out of the fully connected or the sparse case. Hence, one needs to perform numerical simulations in order to investigate the physics of the model.

The ordered version of the ML 4-phasor model has been extensively studied through numerical simulations in [55, 56], where the essential consequences of the FMC on the topology of the interaction graph have been investigated. In particular, the dilution induced by the FMC Eq. (1) has been compared with a random dilution of the same order, revealing important differences between the two cases. The random diluted graph has a homogeneous topology and its phenomenology is compatible with the mean-field solution [37, 57]. On the other hand the inhomogeneities induced by the FMC lead to a graph characterized by a correlated topology and its behaviour significantly differs from the homogeneous mean-field solution because of the onset of phase waves, at least at all simulated $N$. Already in the ordered case, thus, the ML model might display very strong finite size effects. The more so when quenched disordered couplings are considered.

Large sizes are hard to simulate because the mode variables are continuous (complex) numbers and because the total number of interactions grows like $N^3$ with the number $N$ of modes. Because of these effects it has not been possible so far to identify the universality class of the modes. By looking at the specific heat behaviour, it has been observed [39] that for a $O(N)$ dilution having a random homogeneneous or a deterministic topology for the same model makes a great difference in terms of interpolation of the critical properties in the thermodynamic limit. A random homogeneous $O(N)$ dilution of the fully connected network allows to see, already at relatively small sizes, a glass transition of the mean-field kind in the same universality class of the Random Energy Model (REM), which is the reference mean-field model for disordered systems with non-linear interactions. On the other hand the deterministic dilution yields apparently a different result.

To unravel such possible difference here we carefully investigate the universality class of the ML 4-phasor model, providing simulations of systems of large enough sizes, large statistics and, above all, introducing a trick to drastically reduce finite size effects.

After a description of the model in Section II, in Section III we explain the strategy used to reduce the finite size effects due to the heterogeneous FMC dilution. In Section IV we present a simple argument to get the exponent for the finite-size scaling (FSS) regime of the specific heat in the REM and generalize it deriving boundaries for the critical exponents of a generic mean-field universality class. We, then, compare this prediction to the specific

heat behaviour in the equilibrium numerical simulations and, through FSS analysis, we assess that the scaling of the specific heat near the glass transition temperature is compatible with a mean-field theory, which is the main outcome of the present work. Eventually, in Section V we present the behaviour of the overlap probability distribution upon lowering the temperature across the random lasing transition, that turns out to be a glass transition. The trick used to reduce finite-size effects turns out to be useful in identifying more clearly signatures of glassiness.

## II. THE MODE-LOCKED 4-PHASOR MODEL

The ML 4-phasor model has its roots in the quantum theory of the electromagnetic field and matter interaction in an open system. A full account of the derivation of the classical stochastic dynamics from the quantum many-body dynamics of light coupled with matter can be found in [34].

The main point is that by considering laser media where the characteristic time of atomic pump and loss are much shorter than the lifetimes of the resonator modes, the atomic variables, i.e., matter fields, can be removed obtaining non-linear equations for the electromagnetic field alone.

The stochastic differential equation for the time evolution of the modes $a_k$ reads as

$$\frac{da_{k_1}}{dt} = \sum_{\boldsymbol{k}|\text{FMC}(\boldsymbol{k})} g^{(2)}_{k_1 k_2} a_{k_2}$$
$$+ \sum_{\boldsymbol{k}|\text{FMC}(\boldsymbol{k})} g^{(4)}_{k_1 k_2 k_3 k_4} a_{k_2} \bar{a}_{k_3} a_{k_4} + \eta_{k_1}(t), \quad (2)$$

where the expression of the sum over the indices $\boldsymbol{k}$ satisfying a FMC, like (1), will be soon clarified in Eq. (4). The dynamic variables are $a_k(t) = A_k(t)e^{i\phi_k(t)}$, the complex amplitudes of the light modes comprised by the discrete spectrum of the electromagnetic field

$$\boldsymbol{E}(\boldsymbol{r}, t) = \sum_{k=1}^{N} a_k(t)e^{i\omega_k t} \boldsymbol{E}_k(\boldsymbol{r}) + \text{c.c.} \quad (3)$$

where $\boldsymbol{E}_k(\boldsymbol{r})$ is the space-dependent wavefunction of the mode with frequency $\omega_k$. The noise is taken as a white noise $\langle \eta_k(t) \rangle = 0$, $\langle \eta_j(t)\eta_k(t') \rangle = 2T\delta_{jk}\delta(t - t')$, as we will later discuss. The amplitudes $a_k(t)$ are the remnant of the original creation and annihilation operators of the electromagnetic field quantization, which have been degraded to complex numbers in the semiclassical approximation. By *slow* amplitude mode it is meant that the time scale of the amplitude dynamics is larger than the time scale defined by the frequency of the mode, i.e., $\omega_k^{-1}$. Therefore, in the slow amplitude approximation the phases $e^{i\omega_k t}$ can be averaged out, which, in Fourier space, taking the Fourier transform of Eq. (3), implies that $a_k(t) \simeq a_k(t, \omega)\delta(\omega - \omega_k)$. Lasing modes are slow

amplitude modes by definition, since they are characterized by a very narrow linewidth $\gamma$ around their frequency $\omega_k$. The time average of the fast oscillations $e^{i\omega_k t}$ leads to the sum termed FMC in the equation (2). The general expression for $2n$-body interactions reads as

$$\text{FMC}(\boldsymbol{k}) : |\omega_{k_1} - \omega_{k_2} + \cdots + \omega_{k_{2n-1}} - \omega_{k_{2n}}| \lesssim \gamma, \quad (4)$$

of which Eq. (1) is the case $n = 2$. The FMC acts as a selection rule on the modes participating in the interactions.

The linear terms in Eq. (2) yield different contributions possibly depending on cavity gain and losses and atom-field interaction inside the disordered medium. The latter expression is the most relevant one in the dynamics:

$$g^{(2)}_{k_1 k_2} \propto \sqrt{\omega_{k_1}\omega_{k_2}} \sum_{\alpha\beta}^{\{x,y,z\}} \int_V d\boldsymbol{r}\, \epsilon_{\alpha\beta}(\boldsymbol{r})\, E^{\alpha}_{k_1}(\boldsymbol{r})\, E^{\beta}_{k_2}(\boldsymbol{r}), \quad (5)$$

where $\boldsymbol{\epsilon}(\boldsymbol{r})$ is the dielectric permittivity tensor and the integral is extended over the entire volume $V$ of the medium. In particular, the diagonal elements of $g^{(2)}_{k_1 k_2}$ represent the net gain curve of the medium (i.e., the gain reduced by the losses), which plays an important role mainly below the lasing threshold.

The non-linear couplings $g^{(4)}_{k_1 k_2 k_3 k_4}$ are given by the spatial overlap of the electromagnetic mode wavefunctions modulated by a non-linear optical susceptibility $\chi^{(3)}$

$$g^{(4)}_{k_1 k_2 k_3 k_4} \propto \prod_{j=1}^{4} \sqrt{\omega_{k_j}} \sum_{\alpha\beta\gamma\delta}^{\{x,y,z\}} \int_V d\boldsymbol{r}\, \chi^{(3)}_{\alpha\beta\gamma\delta}(\{\omega_{\boldsymbol{k}}\}; \boldsymbol{r})$$
$$\times E^{\alpha}_{k_1}(\boldsymbol{r})\, E^{\beta}_{k_2}(\boldsymbol{r})\, E^{\gamma}_{k_3}(\boldsymbol{r})\, E^{\delta}_{k_4}(\boldsymbol{r}), \quad (6)$$

where, again, the integral is over the whole volume of the medium. In general, both the linear and the non-linear couplings are complex numbers and can be written as [50]

$$g^{(2)}_{k_1 k_2} = G_{k_1 k_2} + iD_{k_1 k_2}, \quad (7)$$
$$g^{(4)}_{k_1 k_2 k_3 k_4} = \Gamma_{k_1 k_2 k_3 k_4} + i\Delta_{k_1 k_2 k_3 k_4}. \quad (8)$$

In the standard laser case the linear couplings are diagonal and the non-linear ones can be safely considered as constant: in this case, $D_k$ is the group velocity dispersion coefficient and $\Delta$ is the self-phase modulation coefficient, responsible for the Kerr effect [40]. In the purely dissipative limit [34, 37], i.e. $D_{k_1 k_2} \ll G_{k_1 k_2}$ and $\Delta_{k_1 k_2 k_3 k_4} \ll \Gamma_{k_1 k_2 k_3 k_4}$, which in standard laser theory corresponds to neglect the group velocity dispersion and the Kerr effect, the dynamics of Eq. (2) becomes a potential differential equation

$$\frac{da_{k_1}}{dt} = -\frac{\partial \mathcal{H}[\boldsymbol{a}]}{\partial \bar{a}_{k_1}(t)} + \eta_{k_1}(t),$$

with a Hamiltonian function given by

$$\mathcal{H} = - \sum_{\boldsymbol{k}|\mathrm{FMC}(\boldsymbol{k})} G_{k_1 k_2} \overline{a}_{k_1} a_{k_2}$$
$$- \sum_{\boldsymbol{k}|\mathrm{FMC}(\boldsymbol{k})} \Gamma_{k_1 k_2 k_3 k_4} \overline{a}_{k_1} a_{k_2} \overline{a}_{k_3} a_{k_4} + \mathrm{c.c.}. \qquad (9)$$

In principle, the noise is correlated, i.e. $\langle \eta_{k_1} \eta_{k_2} \rangle \neq \delta_{k_1 k_2}$. However, it can be diagonalized by changing basis of dynamic variables: the decomposition of resonator modes into a slow amplitude basis is not unique [58] and one can use this freedom to build a basis in which the noise has no correlations. The diagonalization of the noise can be done at the cost of having non-diagonal linear interactions, which is not a real complication in the random laser case, since linear couplings already have off-diagonal contributions accounting for the openness of the cavity.

The laser dynamics is brought to stationarity by gain saturation, a phenomenon connected to the fact that, as the power is kept constant, the emitting atoms periodically decade in lower states saturating the gain of the laser. In the same way the dynamics induced by the Hamiltonian Eq. (9) eventually reaches a stationary regime, when a constraint on the total energy contained in the system is added. This argument was first proposed for standard multimode lasers in Refs [37, 57]. In fact, lasers are strongly out of equilibrium: energy is constantly pumped into the system in order to keep population inversion and stimulated emission, and in the case of cavityless systems also compensate the leakages. However, a stationary regime can be described as if the system is at equilibrium with an effective thermal bath, whose effective temperature (a "photonic" temperature) accounts both for the amount of energy $\mathcal{E} = \epsilon N$ stored into the system because of the external pumping and for the spontaneous emission rate. The latter is proportional to the kinetic energy of the atoms, e. g., to the heat bath temperature $T$. Eventually, the external parameter driving the lasing transition turns out to be [33, 34, 50]

$$T_{\mathrm{photonic}} = \frac{T}{\epsilon^2}. \qquad (10)$$

One can also introduce the pumping rate $\mathcal{P}$ [41] as the inverse of the square root of this ratio:

$$\mathcal{P}^2 = \frac{\epsilon^2}{T} = \frac{1}{T_{\mathrm{photonic}}}.$$

In order to mathematically model the gain saturation, an overall spherical constraint can be imposed on the amplitudes fixing the total optical intensity in the system

$$\sum_{k=1}^{N} |a_k|^2 = \epsilon N. \qquad (11)$$

The precise value of the couplings in the Hamiltonian Eq. (9) requires the knowledge of the spatial wavefunctions of the modes, see Eqs. (5) and (6), which is not

available in random lasers, since they are characterized by a complicated spatial structure of the modes. If, as it apparently occurs in glassy random lasers, modes are spatially extended to wide regions of the optically active compound, each mode is nonlinearly interacting with very many others. We will implement such an "extended modes approximation" [59] in our model, where the only relevant factor in the mode-coupling is the FMC, rather than spatial confinement of light modes. In this case, because of thermodynamic convergence, each coupling coefficient will be smaller and smaller as the number of modes increases.

In principle, all couplings involving the same mode will be correlated. However, because of the spatially etherogeneous optical nonlinear susceptibility in (6) and the fact that each coupling coefficient vanishes as $N$ increases, the role of correlation will be qualitatively negligible as far as the system displays enough modes. For this reason, the couplings will be taken as independent Gaussian random variables in the present work:

$$\mathcal{P}(J_{k_1 \cdots k_p}) = \frac{1}{\sqrt{2\pi\sigma_p^2}} \exp\left\{ -\frac{J_{k_1 \cdots k_p}^2}{2\sigma_p^2} \right\}, \qquad (12)$$

with $p = 2, 4$ and $\sigma_p^2 \sim N^{2-p}$ to ensure the extensivity of the Hamiltonian and where some rescaling of the modes and coefficients ($g \to J$) has been performed [34, 41]. Eventually, the stationary properties of the system can be described by a model whose Hamiltonian is

$$\mathcal{H}[\boldsymbol{a}] = \mathcal{H}_2[\boldsymbol{a}] + \mathcal{H}_4[\boldsymbol{a}], \qquad (13)$$

where

$$\mathcal{H}_2[\boldsymbol{a}] = - \sum_{\boldsymbol{k}|\mathrm{FMC}(\boldsymbol{k})} J_{k_1 k_2} \overline{a}_{k_1} a_{k_2} + \mathrm{c.c.}$$
$$\mathcal{H}_4[\boldsymbol{a}] = - \sum_{\boldsymbol{k}|\mathrm{FMC}(\boldsymbol{k})} J_{k_1 k_2 k_3 k_4} \overline{a}_{k_1} a_{k_2} \overline{a}_{k_3} a_{k_4} + \mathrm{c.c.} \qquad (14)$$

As mentioned in section II when introducing the dissipative limit we will consider the $J$'s as real parameters, without loss of generality. The effective distribution for the phasor configuration $\boldsymbol{a} = \{a_1, \ldots, a_N\}$ will, eventually, be

$$\mathcal{P}[\boldsymbol{a}] \propto e^{-\beta\mathcal{H}[\boldsymbol{a}]} \delta\left( \epsilon N - \sum_{k=1}^{N} |a_k|^2 \right), \qquad (15)$$

where $\beta$ is the inverse temperature.

## III. FREQUENCY MATCHING CONDITION WITHOUT EDGE-BAND MODES

The FMC Eq. (4) is the most peculiar aspect of the ML 4-phasor model, since it defines the topology of the interaction network. The full inclusion of the FMC in

the study of the model has not been achieved analytically yet, given the difficulty of the problem. The analytical solution of the ML (2+4)-phasor model has been derived only in the *narrow bandwidth* approximation, in which the interaction network is a fully connected graph. In this approximation, the typical bandwidth $\gamma$ of the modes is of the order of the spectrum bandwidth $\Delta\omega$ and the FMC is satisfied by all the modes. To include the FMC means to go beyond the fully connected case, which requires the development of new techniques with respect to standard mean-field methods. However, numerical simulations can yield important insights on the nature of the model.

Besides being relevant from a purely theoretical point of view, dealing with the FMC is also important in order to provide a realistic description of random lasers. The FMC is, indeed, responsible for mode-locking [40] at the lasing transition. Mode-locking is the regime under which a standard multimode laser generates ultrashort pulses, due to the formation of phase waves of nontrivial slope [55]. In random lasers the mode couplings are non-perturbatively disordered disrupting the onset of a laser pulse. However, the underlying phenomenon of phase locking might still be present, though as a self-starting phenomenon [41] rather than induced by ad hoc devices as in standard mode-locking lasers [40].

Frequencies are, in principle, not equispaced in random lasers and their convolution would prevent the onset of pulses in time even in presence of unfrustrated couplings. Since, however, because of quenched disorder in the couplings no pulse is there notwithstanding the distribution of the mode frequencies, we consider here for simplicity a *frequency-comb* distribution:

$$\omega_k = \omega_1 + (k-1)\delta\omega \qquad k = 1, ..., N \qquad (16)$$

with $\gamma \ll \delta\omega$ and the central frequency given by $\omega_0 \simeq \omega_1 + N\delta\omega/2$.

We note that in this case the linear term of the complete Hamiltonian Eq. (13) is diagonal. If we assume that the diagonal part of the pairwise couplings does not depend on the modes, together with the spherical constraint Eq. (11), this term is an irrelevant additive constant. The diagonal part of the linear contribution to the Hamiltonian physically represents the gain profile of the optical random medium (possibly becoming a random laser at high pumping). As a working hypothesis we are assuming a uniform gain profile over the whole spectrum. For the numerical simulations of this work, then, we have sampled configurations of the light modes according to the equilibrium probability distribution in Eq. (15) with $\mathcal{H} = \mathcal{H}_4$ the four-body term defined in Eq. (14). Due to FMC the only non zero contribution to $\mathcal{H}_4[\boldsymbol{a}]$ comes from the frequencies which fulfill the constraint (1). More notably, with Eq. (16) the condition (1) on the frequencies can be mapped into a condition on the indices of the interaction graph

$$|k_1 - k_2 + k_3 - k_4| = 0. \qquad (17)$$

The FMC in Eq. (17) tends to cut order $O(N)$ interacting quadruplets with respect to the complete graph [51]. Therefore, the total amount of couplings in the network is $O(N^3)$ and each phasor spin in the system will be interacting in $O(N^2)$ quadruplets. Though diluted with respect to the complete graph, the network is still dense.

The FMC also introduces non-linear correlations in the interactions affecting the topology of the interaction network. Modes with more similar frequencies are connected by a higher number of quadruplets and, consequently, they are effectively more coupled. As a consequence, modes whose frequencies are at the center of the spectrum ($\omega \simeq \omega_0$) tend to interact more than modes whose frequencies are at the boundaries ($\omega \simeq \omega_1$ or $\omega \simeq \omega_1 + N\delta\omega$). This can be clearly seen in the emission spectrum $I_k$ resulting from the numerical simulations for a given fixed instance of the disorder, which is shown in Fig. 1. Data are obtained using the Monte Carlo Exchange algorithm, also known as Parallel Tempering, allowing to reach equilibrium on relatively short simulation times. All observables analyzed here are drawn from configurations at equilibrium. All details about the numerical simulation algorithm and the computation of the equilibrium thermal averages are discussed in App. A.

Let us briefly comment on the relationship between the physical intensities $I_k$ and the complex amplitude variables of the simulated model (14). In real experiments the heat bath temperature $T$ is typically kept fixed (there are exceptions like, e.g., in Ref. [60]) and the overall system energy $\mathcal{E} = \epsilon N$ is varied by tuning the pumping power. In our simulations, $\epsilon$ is fixed and kept equal to one in the spherical constraint, $\sum_k |a_k|^2 = \sum_k A_k^2 = N$, whereas $T$ is varied. Therefore, according to Eq. (10) a change in the pumping rate $\mathcal{P}$ because of a shift in the energy $\epsilon$ pumped into the system corresponds to a shift of $1/\sqrt{T}$. If we rescale the intensity of the mode $k$ as

$$I_k = \frac{A_k^2}{\sqrt{T}} \qquad (18)$$

we have $\sum_k I_k = N/\sqrt{T} = N\epsilon$, as in Eq. (11).

One of the most relevant features of the intensity spectrum shown in Fig. 1 is that it becomes more and more structured and heterogeneous upon decreasing the temperature [61]. Another interesting feature of the spectrum is the central band narrowing, akin to the spectra of true experimental realizations of random lasers [2, 3]. This is a consequence of the fact that band-edge modes are less interacting and, as far as numerical simulations are concerned, is one of the reasons why this sort of simulations are plagued by strong finite-size effects.

In order to reduce these finite-size effects we have imposed periodic boundary conditions on the frequencies when filtering couplings with the FMC condition. This has the effect of eliminating band-edge modes, or, equivalently, it is like considering only modes at the center of the spectrum in a much larger system. The periodic boundary conditions of the frequencies are obtained in

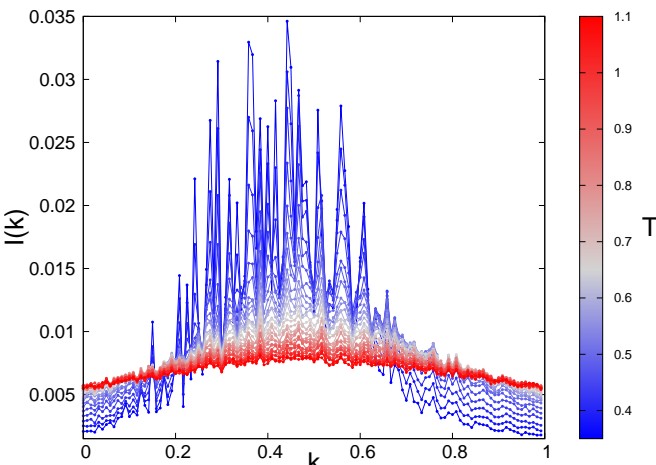

FIG. 1. Intensity spectrum $I_k$, Eq. (18), for a single realization of quenched disorder of the ML 4-phasor model with free boundary conditions on the frequencies and $N = 120$ modes. Temperature $T \in [0.7, 1.45]$ (color map on the vertical bar). Notice the narrowing of the central part of the spectrum, because of FMC and the onset of isolated spikes as $T$ decreases, signaling breaking of intensity equipartition. The pattern of the peaks is disordered and strongly depends on the random sample and on the single dynamic history.

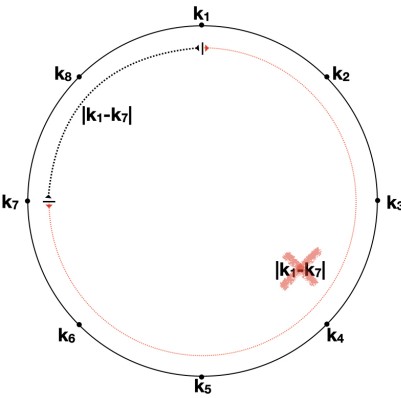

FIG. 2. Periodic boundary conditions on the mode frequency indexes for the frequency matching condition.

practice by representing the frequency indices as variables on a ring, see Fig. 2, and taking their distance as the smallest one between any two of them:

$$|k_a - k_b| = \begin{cases} |k_a - k_b| & \text{if } |k_a - k_b| \le \left[\frac{N}{2}\right] \\ N - |k_a - k_b| & \text{if } |k_a - k_b| \ge \left[\frac{N}{2}\right] \end{cases}, \tag{19}$$

where $[n]$ is the integer part of $n$. From now on we refer to the version of the ML 4-phasor model with periodic boundary condition on the frequencies as PBC, whereas the original one, with free boundary conditions will be termed FBC.

In Fig. 3 we show the emission spectrum at equilibrium for a single instance of disorder for the ML 4-phasor

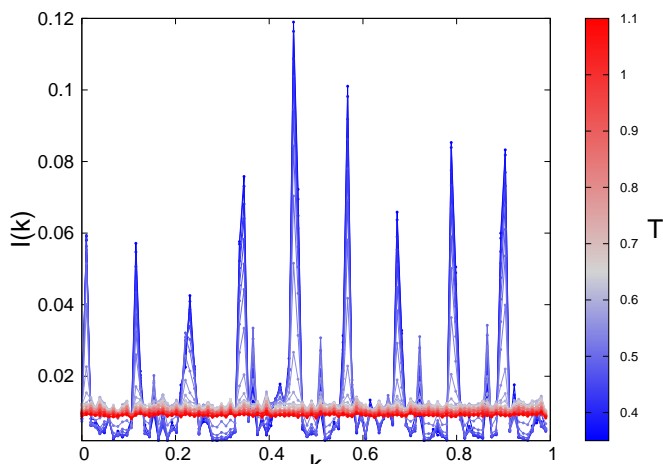

FIG. 3. Intensity spectrum $I_k$ Eq. (18) for a single realization of quenched disorder of the ML 4-phasor model with periodic boundary conditions on the frequencies and $N = 104$. The spectrum is normalized and the modes $k$ are divided by $N$. Temperature $T \in [0.35, 1.1]$ (color map on the right hand vertical bar). Notice the loss of the spectrum curvature, due to periodic boundary conditions on the FMC and the persistence of the isolated peaks.

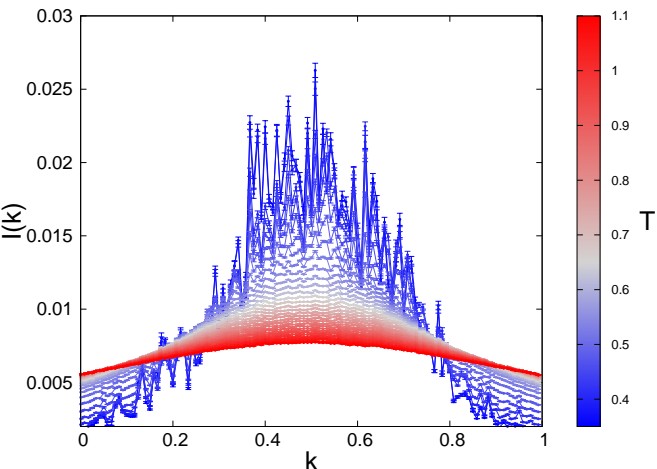

FIG. 4. Intensity spectrum $I_k$ Eq. (18) of the ML 4-phasor model with free boundary conditions on the frequencies and $N = 120$ averaged over $N_s = 75$ instances of quenched disorder. Temperature $T \in [0.7, 1.5]$ (color map on the vertical bar). Averaging over disorder smoothens the spectra.

model with PBC for the frequencies. The most relevant difference with respect to the case of FBC is the complete absence of narrowing in the spectrum, which corresponds to the absence of band-edge modes: all modes interact with identical probability with the rest of the system.

In Figs. 4 and 5 we also show the FBC and PBC spectra averaged over roughly a hundred instances of disorder. In Fig. 4 one can observe the typical narrowing occurring in random lasers [2, 3, 5, 28] as the pumping energy increases. Fig. 5 displays flat spectra in the low pumping regime, and homogeneously distributed random

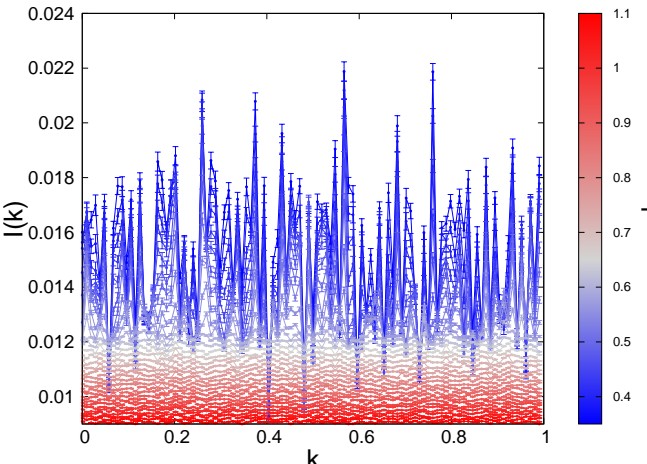

FIG. 5. Intensity spectrum $I_k$ Eq. (18) averaged over $N_s = 80$ samples of the model with periodic boundary conditions on the frequencies and $N = 104$ modes. Temperature $T \in [0.35, 1.1]$ (color map on the vertical bar).

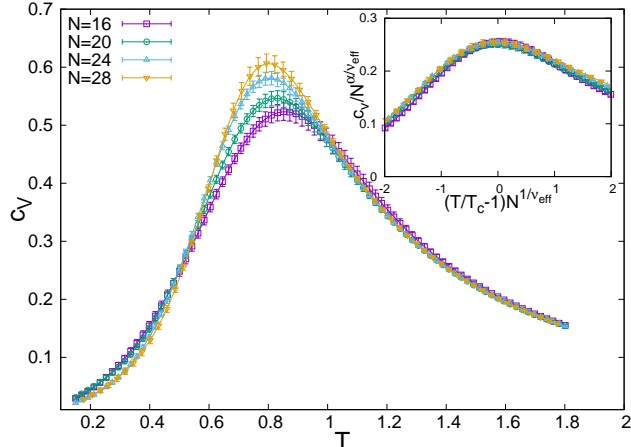

FIG. 6. Specific heat of the Random Energy Model. Different colors represent different simulated sizes at finite sizes $N = 16, 20, 24, 28$. (Inset) Specific heat rescaled by $N^{\alpha/\nu_{\text{eff}}}$, $\nu_{\text{eff}} \equiv 2\beta + \gamma$, as a function of $\tau N^{1/\nu_{\text{eff}}}$. The best data collapse has been obtained with $\alpha = 0.52(7)$ and $2\beta + \gamma = 1.9(2)$

.

resonances in the high pumping regime. They look like the central part of the spectra of Fig. 4.

## IV.  UNIVERSALITY CLASS

In a $\phi^4$ mean-field theory (a Landau theory) the critical exponents characterizing the universality class are $\beta = 1/2$ for the order parameter $\langle\phi\rangle$, $\gamma = 1$ for the susceptibility $\chi$ and $\nu = 1/2$ for the correlation length. They satisfy the hyperscaling relation $2\beta + \gamma = \nu d$, holding for all dimensions $d \leq d_{\text{uc}}$, the upper critical dimension, that is $d_{\text{uc}} = 4$ in a $\phi^4$ model. As an instance, this is the universality class of the Random Energy Model (REM), a reference simplified model for the glass transition. This is also the universality class of the mean-field 4-phasor model representing a random laser in the so-called narrow-band approximation, both in a fully connected interaction network, where the solution can be analytically computed [33] and in a uniformly randomly diluted version of the model, analyzed by means of equilibrium Monte Carlo simulations in Ref. [39].

Moving to the more realistic random laser models, where the basic ingredient for mode-locking, the frequency matching condition (1) is implemented, it is more difficult to understand whether the universality class remains the same. In Ref. [39] an estimate of the value of the critical exponent $\nu_{\text{eff}} \equiv 2\beta + \gamma \simeq 2/3$ was provided for the mode-locked random laser model. This result is quite different from the value $2\beta + \gamma = 2$ which characterizes the REM model, even if we consider its numerical finite-size scaling analysis.

As an instance, the REM specific heat behaviour for small sizes $N = 16, 20, 24, 28$ is reported in Fig. 6. Details about the numerical technique used are given in App. B. Even though the simulated $N$ are not very large,

from the interpolation of the $c_V(T)$ peaks it turns out that $\nu_{\text{eff}} = 2\beta + \gamma = 1.9 \pm 0.2$. Strong finite size effects are there, as one can observe from the estimate of the $\alpha$ exponent, displaying a value $\alpha = 0.52 \pm 0.07$, rather different from the mean-field exponent $\alpha = 0$. Because of preasymptotic effects, indeed, the scaling relation $2\beta + \gamma + \alpha = 2$ (independent from the system dimension) appears to be violated.

### A.  Mean-field exponent

The exponent value $\nu_{\text{eff}} = 2\beta + \gamma = 2$ can be derived through a simple argument, which does not require any specific knowledge of the model and can be easily generalized. Let us consider a mean-field theory described by the Ginzburg-Landau potential

$$V(\phi) = \frac{1}{2}\tau\phi^2 + \frac{g}{4!}\phi^4, \tag{20}$$

where $\phi$ represents the global order parameter of the transition and $\tau$ is the reduced temperature $\tau = T/T_c - 1$. This is the standard paradigm of a second order phase transition, as a glass transition is known to be, as far as the thermodynamic potential and its derivatives (including the specific heat) are concerned [62]. The critical behaviour of a susceptibility in a second order transition is related to the fluctuations of the order parameter,

$$\delta\phi^2 = \langle\phi^2\rangle - \langle\phi\rangle^2 \propto \chi \sim \frac{1}{N|\tau|^\gamma}, \tag{21}$$

where the average is assumed to be taken with respect to the probability distribution

$$P(\phi) \propto e^{-NV(\phi)} \tag{22}$$

and $N$ represents the size of the system. When $\tau \gtrsim 0$ the effective potential is well approximated by $V(\phi) \simeq \frac{1}{2}\tau\phi^2$, at least for values of the field close enough to the minimum $\phi = 0$. The partition function that normalizes the probability distribution can be then computed through a simple Gaussian integration

$$Z \simeq \int d\phi \; e^{-\frac{N\tau}{2}\phi^2} \sim \frac{1}{\sqrt{N\tau}}, \qquad \tau \gtrsim 0. \qquad (23)$$

Hence, in this regime the fluctuations of the order parameter centered around the minimum $\phi = 0$ are given by the variance of the Gaussian distribution.

On the other hand, when $\tau \lesssim 0$ the quartic term of the potential becomes relevant and cannot be neglected. In this regime, the fluctuations of the order parameter are centered around one of the two symmetric minima of the potential (20), namely $\phi_\pm = \pm\phi^*$, depending on the initial conditions. Since we are interested in matching the fluctuations above and below the critical temperature, we assume the temperature to be sufficiently close to $T_c$ in order for the amplitude of the fluctuations to be of the order of the distance from the origin

$$\delta\phi^2 \simeq (\phi^*)^2. \qquad (24)$$

The minima $\phi_\pm$ can be easily determined according to the saddle-point approximation of the partition function

$$Z = \int d\phi \; e^{-NV(\phi)} \simeq e^{-NV(\phi^*)}, \qquad (25)$$

where $\phi^*$, solution to the saddle point equation,

$$\left.\frac{dV(\phi)}{d\phi}\right|_{\phi^*} = 0$$

turns out to be

$$\phi^* = \sqrt{\frac{6|\tau|}{g}}. \qquad (26)$$

We have therefore an estimate of the fluctuations on the two sides of the critical point, respectively

$$\delta\phi^2_{T>T_c} \sim \frac{1}{N\tau} \qquad (27)$$

$$\delta\phi^2_{T<T_c} \sim |\tau|/g. \qquad (28)$$

The dependence on $N$ of the scaling regime can be obtained by matching the order of magnitude of the fluctuations above and below the critical temperature:

$$\delta\phi^2_{T>T_c} \sim \delta\phi^2_{T<T_c} \quad \Longrightarrow \quad |\tau| \sim \frac{1}{N^{1/2}}. \qquad (29)$$

Let us recognize that Eq. (27) is the susceptibility, cf. Eq. (21), whereas Eq. (28) is the scaling of the square of the order paramater $\langle\phi\rangle = \phi^*$, which is scaling as $\phi^* \sim |\tau|^\beta$. Therefore,

$$|\tau| \sim \frac{1}{N^{1/(2\beta+\gamma)}}.$$

In the mean-field $\phi^4$ theory $2\beta + \gamma = 2$. Since the upper critical dimension is $d_{uc} = 4$ this corresponds to $\nu d_{uc} \equiv \nu_{eff} = 2$, i.e., $\nu = 1/2$ for the mean-field critical correlation length exponent.

The previous argument can be straightforwardly extended to a more general mean-field potential, in order to obtain a range of values for the critical exponents compatible with mean-filed theories. Let us consider the potential

$$V(\phi) = \frac{1}{2}\tau\phi^2 + \frac{g}{n!}\phi^n, \qquad (30)$$

The fluctuations of the order parameter above the critical temperature are the same as in the case $n = 4$. By imposing Eq. (24) and solving the saddle-point equation for $\phi^*$, one finds that in the generic case

$$\phi^* = \left[\frac{(n-1)! \; |\tau|}{g}\right]^{\frac{1}{n-2}} \sim |\tau|^\beta, \qquad (31)$$

yielding

$$\beta = \frac{1}{n-2}.$$

Therefore, the matching of the amplitude of the fluctuations above, cf. Eq. (27), and below $T_c$, i.e.,

$$\delta\phi^2_{T<T_c} \sim (|\tau|/g)^{\frac{2}{(n-2)}}, \qquad (32)$$

leads to

$$\delta\phi^2_{T>T_c} \sim \delta\phi^2_{T<T_c} \quad \Longrightarrow \quad |\tau| \sim \frac{1}{N^{\frac{n-2}{n}}} = \frac{1}{N^{\frac{1}{2\beta+\gamma}}}. \quad (33)$$

This result implies that, in order to be compatible with mean-field theory, the values of $\nu_{eff} = 2\beta+\gamma = n/(n-2)$ must fall in an interval defined by taking $n = 4$ and $n \to \infty$ in the previous expression. Eventually, the critical exponent for the scaling of the specific heat width in a generic mean-field theory must take value in the interval

$$1 \leq \nu_{eff} \leq 2. \qquad (34)$$

Given the specific theory $\phi^n$ and its upper critical dimension $d_{uc}(n)$, the critical mean-field exponent $\nu$ is equal to $\nu = \nu_{eff}/d_{uc}(n)$.

In the model under consideration, though, we have a dense (though not fully connected) interaction network and we do not have a reference $d$-dimensional lattice underneath, such that a scaling relation of the number of modes to a characteristic length can be set, as, for instance $N = L^d$ in a $d$-dimensional hypercubic lattice. Our analysis will, therefore, be limited to the estimate of the exponents $\alpha$, $\beta$ and $\gamma$.

It is also worth noting that the previous argument is exact only in the large-$N$ limit, where the saddle-point approximation holds. It is therefore likely that numerical simulations at finite $N$ display finite-size effects which deviate from the above estimate. In particular for dense

models as the one we are studying it is difficult to access higher values of $N$ because the number of interacting quadruplets increases as $N^3$ and the computational cost of the simulations is the one of a Non-deterministic Polynomial Complete problem. For help decreasing the finite size effects we have exploited the alternative strategy discussed in Sec. III, whose results are presented in the following subsection.

### B. Finite-size scaling analysis: numerical results

We perform a finite-size scaling study of the specific heat obtained from our numerical simulations, in order to determine the value of the critical exponents $\alpha$ and $\nu_{\mathrm{eff}}$. Let us define the absolute value of the reduced temperature $t = |T/T_c - 1|$. In general, the basic assumption of the FSS Ansatz [63, 64] is that the finite-size behaviour of an observable $Y_N$ in a system of size $N$ is governed by the ratio between the correlation length $\xi_\infty$ of the infinite system and the size $N$. In the thermodynamic limit near the critical point the observable $Y$ scales like

$$Y_\infty(T) \approx A t^{-\psi}.$$

The correlation length $\xi_\infty$ scales like

$$\xi_\infty(T) \approx \xi_0 t^{-\nu}. \tag{35}$$

The scaling hypothesis can, then, be written as

$$Y_N(T) = N^{\frac{\omega}{d}} f_Y \left( \frac{\xi_\infty^d}{N} \right), \tag{36}$$

where $\omega$ is the critical exponent for the scaling of the peak of the observable and $f_Y$ is a dimensionless function that depends on the observable $Y$. The function $f_Y$ is such that in the limit $N \to \infty$ one recovers the scaling law $Y_\infty(T) \approx A t^{-\psi}$, an hence, by using (35), $\omega = \psi/\nu$ [63]. Therefore combining Eqs. (35) and (36) the scaling relation becomes

$$Y_N(T) = N^{\frac{\psi}{\nu d}} \hat{f}_Y \left( N^{\frac{1}{\nu d}} t_N \right) = N^{\frac{\psi}{\nu_{\mathrm{eff}}}} \hat{f}_Y \left( N^{\frac{1}{\nu_{\mathrm{eff}}}} t_N \right), \tag{37}$$

where $t_N = |T/T_c(N) - 1|$, $T_c(N)$ is the finite-size critical temperature and $\hat{f}_Y$ is another scaling function. In the case of the specific heat, the previous finite-size scaling law takes the following form

$$c_{V_N}(T) = N^{\frac{\alpha}{\nu_{\mathrm{eff}}}} \hat{f}_{C_{V_N}} \left( N^{\frac{1}{\nu_{\mathrm{eff}}}} t_N \right), \tag{38}$$

where $\alpha$ denotes the critical exponent of the specific heat peak divergence. Since the dimensionless function $\hat{f}$ is scaling invariant, if one uses the correct values of the exponents $\alpha$ and $\nu_{\mathrm{eff}}$, the curves $c_{V_N}(T)/N^{\alpha/\nu_{\mathrm{eff}}}$ for different values of $N$ should collapse on the same curve.

In order to get the two exponents $\alpha$ and $\nu_{\mathrm{eff}}$ from our numerical data we follow the scaling method of Refs.

[65, 66], whose details are reported in App. C. For each size $N$ the specific heat is measured by calculating the equilibrium energy fluctuations at each temperature $T$ and then averaging over disorder instances

$$c_{V_N} = \frac{1}{N} \frac{\overline{\langle E^2 \rangle - \langle E \rangle^2}}{T^2}, \tag{39}$$

where $\langle \dots \rangle$ represents the thermal average and $\overline{[\dots]}$ represents the average over disorder, see App. C.

For the systems with FBC, the specific heat behaviour as a function of temperature is shown in the main panel of Fig. 7 for different sizes. By a quadratic fit of the peaks of the specific heat (at $T_c(N)$), the critical temperature is estimated to be $T_c = 0.86(3)$ in the thermodynamic limit, as interpolated in Appendix C. In the inset of Fig. 7 data are collapsed using the exponents $\alpha$ and $\nu_{\mathrm{eff}}$ obtained from the FFS analysis reported in App. C:

$$\text{FBC:} \quad \alpha = 0.48 \pm 0.05, \quad 1/\nu_{\mathrm{eff}} = 1.1 \pm 0.1. \tag{40}$$

In order to perform the FSS analysis we have used the temperatures reported in Table I.

With respect to the estimate $1/\nu_{\mathrm{eff}} \simeq 1.5$ found in [39], a much larger statistics allows now to find an estimate of $2\beta + \gamma$ closer to the mean-field threshold and suggesting that deviations from mean-field theory might be due to pre-asymptotic effects in $N$. The confirmation that this is, indeed, the origin of the anomalous value previously found for $2\beta + \gamma$ comes from the analysis with frequency PBC, devised to partially circumvent finite size corrections.

The specific heat for systems whose frequencies obeys PBC are displayed in Fig. 8. In the main panel we show the raw data. Analyzing the scaling of the peak the critical temperature $T_c = 0.61(3)$ has been determined. In the inset of Fig. 8 we show the collapsed data with the values of exponents derived with the FSS method reported in App. C

$$\text{PBC:} \quad \alpha = 0.27 \pm 0.05, \quad 1/\nu_{\mathrm{eff}} = 0.86 \pm 0.14. \tag{41}$$

With PBC we find an estimate inside the interval (34) for a mean-field universality class. Therefore, up to the limits of our analysis, despite being possibly still of a

| | FBC | | | PBC | | |
|---|---|---|---|---|---|---|
| $N_4$ | $N$ | $T_c$ | $\Delta T_c$ | $N$ | $T_c$ | $\Delta T_c$ |
| $2^8$ | 18 | 0.55 | 0.04 | - | - | - |
| $2^9$ | - | - | - | 18 | 0.42 | 0.02 |
| $2^{11}$ | 32 | 0.63 | 0.025 | 28 | 0.49 | 0.02 |
| $2^{13}$ | 48 | 0.69 | 0.02 | 42 | 0.52 | 0.02 |
| $2^{14}$ | 62 | 0.75 | 0.03 | 54 | 0.55 | 0.03 |
| $2^{15}$ | - | - | - | 66 | 0.56 | 0.04 |
| $2^{16}$ | 96 | 0.8 | 0.07 | 82 | 0.56 | 0.05 |
| $2^{17}$ | 120 | 0.83 | 0.09 | - | - | - |

TABLE I. Values of the critical temperatures for the ML 4-phasor model with fixed and periodic boundary conditions.

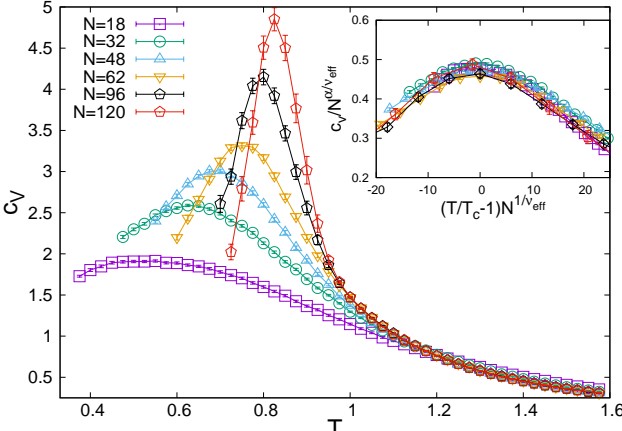

FIG. 7. Specific heat $c_{V_N}$, (39), for the ML 4-phasor model with free boundary conditions on the frequencies as a function of $T$. Different curves represent different simulated sizes of the system. The simulated sizes are $N = 18, 32, 48, 62, 96, 120$. (Inset) Specific heat scaled by $N^{\alpha/\nu_{\text{eff}}}$ as a function of $\tau N^{1/\nu_{\text{eff}}}$, with $\alpha = 0.48(5)$, $\nu_{\text{eff}} = 2\beta + \gamma = 0.91(8)$.

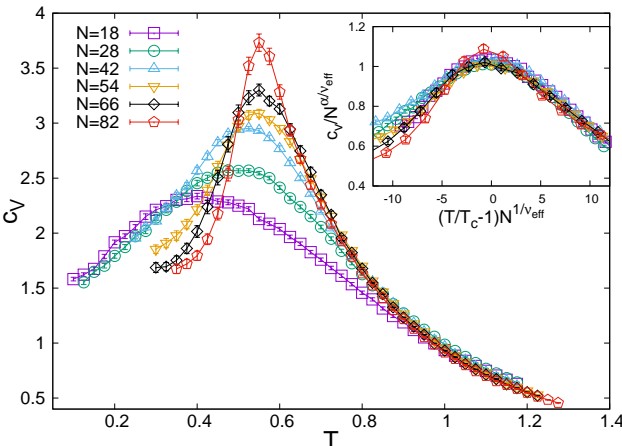

FIG. 8. Specific heat $c_{V_N}$, (39), for the ML 4-phasor model with periodic boundary conditions on the frequencies as a function of $T$. Different curves represent different simulated sizes of the system. The simulated sizes are $N = 18, 28, 42, 54, 66, 104$. (Inset) Specific heat scaled by $N^{\alpha/\nu_{\text{eff}}}$ as a function of $\tau N^{1/\nu_{\text{eff}}}$, with $\alpha = 0.27(5)$, $\nu_{\text{eff}} = 2\beta + \gamma = 1.2(2)$.

different universality class with respect to the REM, for which $2\beta + \gamma = 2$, we observe that the glass transition of the ML 4-phasor model is compatible with a mean-field transition.

## V. THE GLASS TRANSITION

The mean-field paradigm used to describe the thermodynamics of the glass transition is the *random first order transition* (RFOT), which is a mixed-order ergodicity breaking transition [67–70]. The ML 4-phasor model

displays the features of a RFOT. The second order nature of the transition is exhibited by the specific heat anomaly studied in the previous section. In this section we aim to complete the study of the glass transition of the ML 4-phasor model, by focusing on its first order nature, which is represented by the discontinuity of the order parameter.

The order parameter for the glass transition is the overlap probability distribution $P(q)$ [42]. In models with continuous variables, the $P(q)$ is expected to be a distribution with a single peak in $q = 0$ in the high temperature phase and to develop side peaks, as well, in the low temperature glass phase. At finite $N$, of course, exact Dirac delta peaks in the $P(q)$ appear as a smoothen function of $q$ due to strong finite-size effects.

Overlaps are defined as scalar products among phasor configurations of independent replicas of the system with the same quenched disorder. In the present case the relevant overlap for the transition turns out to be [34, 50, 71]

$$
\begin{aligned}
q_{\alpha\beta} &= \frac{1}{N}\text{Re}\ \sum_{i=1}^{N} \bar{a}_k^\alpha a_k^\beta \\
&= \frac{1}{N}\sum_{i=1}^{N} A_k^\alpha A_k^\beta \cos(\phi_k^\alpha - \phi_k^\beta),
\end{aligned}
\tag{42}
$$

where $\alpha$ and $\beta$ are replica indexes. Since replica overlaps measure the similarity between glassy states of the system, their distribution gives information about the structure of the phase space.

The protocol used in numerical simulations to measure the overlaps corresponds to the definition of replicas as independent copies of the system with the same quenched disorder. For each sample, i.e., each realization of disorder, we run dynamics independently for $N_{\text{rep}}$ replicas of the system, starting from randomly chosen initial phasor configurations. In this way, replicas explore different regions of the same phase space and may thermalize in configurations belonging to apart states. To study the behavior of the $P_J(q)$ we choose $N_{\text{rep}} = 4$, so that at any measurement time six values of the overlap are available $q_{\alpha\beta} = \{q_{01}, q_{02}, q_{03}, q_{12}, q_{13}, q_{23}\}$. In order to accumulate statistics, we measure the value of $q_{\alpha\beta}$ using $\mathcal{N}$ equilibrium, time uncorrelated, configurations of replicas at the same iteration of the simulated dynamics. Hence, for each disordered sample the $P_J(q)$ histograms are built with $\mathcal{N} \times N_{\text{rep}}(N_{\text{rep}} - 1)/2$ values of the overlap. The number of configurations $\mathcal{N}$ actually used from our data can be evinced from tables II-III in appendix A, in which the last half of the simulated Monte Carlo steps are surely thermalized and the correlation time was estimated to be $2^8$ Monte Carlo steps. Eventually, for each realization of the quenched random couplings we have $\mathcal{N} = 2^{10} - 2^{12}$, depending on the size.

The overlap distribution functions $P_J(q)$ are computed as the normalized histograms of the overlaps for each one of the samples. This has been done for each simulated size of the ML 4-phasor model with both FBC and PBC.

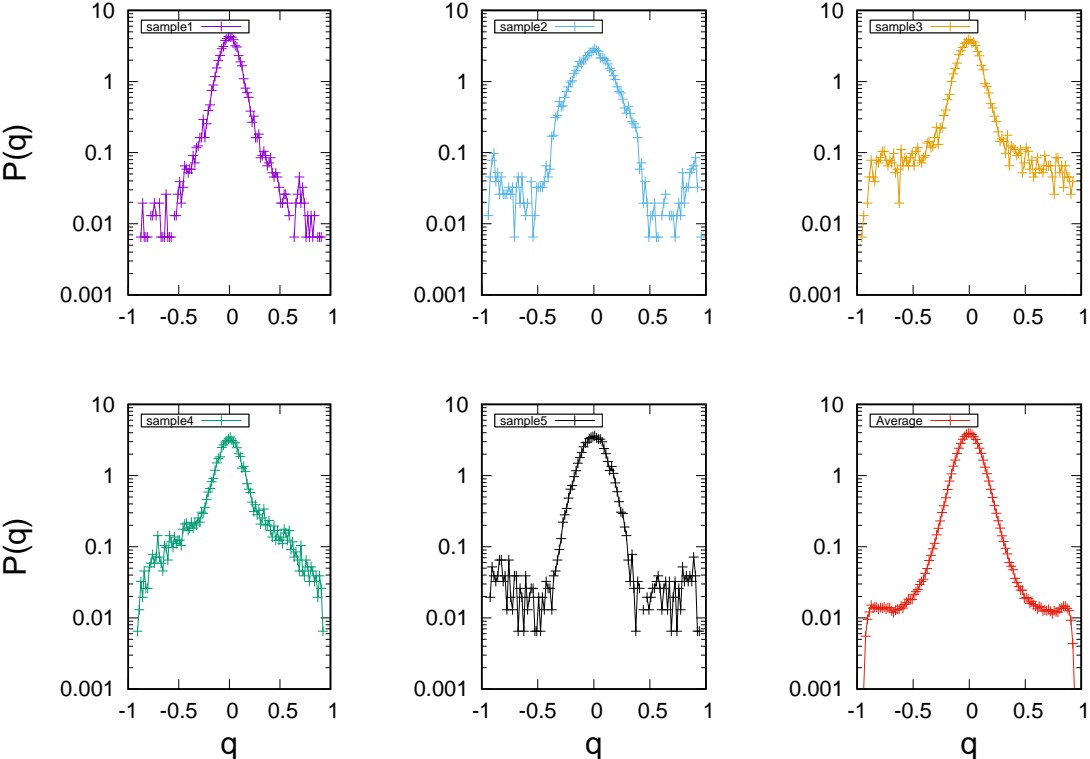

FIG. 9. Overlap distributions for five instances of disorder and for the average over all instances at $T = 0.25 \simeq 0.45\ T_c$. Simulation size $N = 54$ of the ML 4-phasor model with periodic boundary conditions on the frequencies. Notice how a distribution belonging to a sample significantly differs from the others: the relevant quantity for thermodynamics is the overlap distribution averaged over all samples.

In Fig. 9 we present the overlap distributions for five samples at the temperature $T = 0.25 \simeq 0.45 T_c$ of the size $N = 54$ of the ML 4-phasor model with PBC, together with the overlap distribution averaged over 100 samples. Given the fluctuations of $P_J(q)$ among the different samples, it is clear that the only physical quantity to be consider in order to assess the glass transition is the averaged $P(q) \equiv \overline{P_J(q)}$.

This is particularly important in the case of the overlap distribution function, since, contrarily to the other thermodynamic observables, it is not a self-averaging quantity [42], i.e., the average $P(q)$ cannot be reached simply by increasing the size of the system over which a single sample $P_J(q)$ is built, but only by averaging over disorder. In Fig. 10 and Fig. 11 the average overlap distribution function of the ML 4-phasor model with FBC and PBC are, respectively, reported for the whole simulated temperature range in a system with $N = 62$ spins. The reduction of the finite-size effects obtained by using periodic boundary conditions in the choice of interacting modes leads to display $P(q)$ with more distinct secondary peaks in the case of the ML 4-phasor model with PBC.

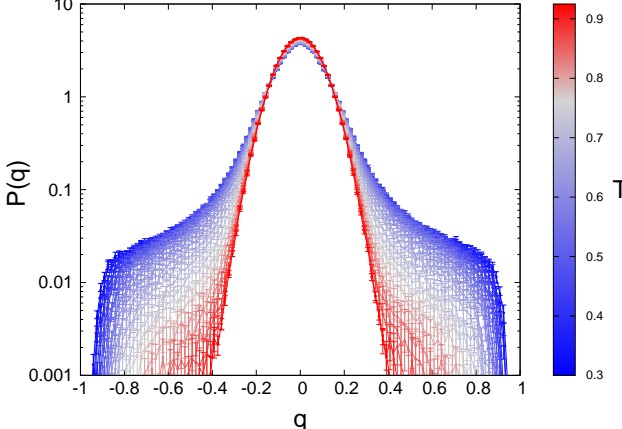

FIG. 10. Parisi overlap distribution for the size $N = 62$ of the ML 4-phasor model with free boundary conditions on the frequencies. The distribution is averaged over $N_s = 100$ instances of disorder. Temperature $T \in [0.3, 0.9]$ (color map on the vertical bar). The blue curve corresponding to the lowest temperature is at $T \simeq 0.4T_c$, with $T_c = 0.86(3)$.

## VI. DISCUSSION AND CONCLUSIONS

In the present work we have simulated the equilibrium dynamics of a leading model for a glassy random

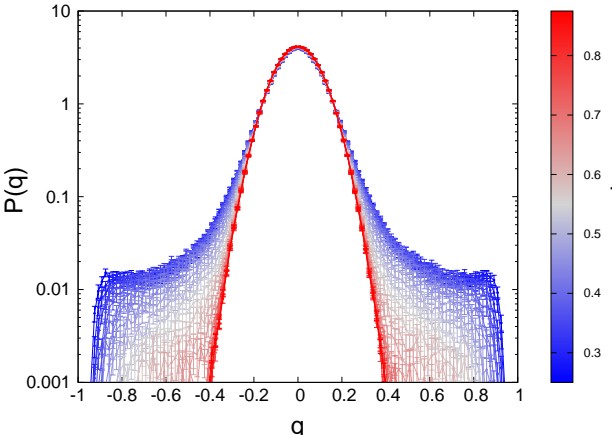

FIG. 11. Parisi overlap distribution for the size $N = 54$ of the ML 4-phasor model with with periodic boundary conditions on the frequencies. The distribution is averaged over $N_s = 100$ instances of disorder. Temperature $T \in [0.3, 0.85]$ (color map on the right hand vertical bar). The blue curve corresponding to the lowest temperature is at $T \simeq 0.45 T_c$, with $T_c = 0.61(3)$. Notice that here the overlap distribution has more pronounced secondary peaks with respect to the overlap distribution in Fig. 10.

laser, that is, a random laser displaying Replica Symmetry Breaking: the mode-locked (ML) 4-phasor model.

Carefully studying the critical behavior of the specific heat, performing a finite size scaling analysis we have estimated the critical exponents. The critical exponents turn out to yield $2\beta + \gamma = \nu_{eff} = 0.91(8)$, slightly below the threshold of 1 for mean-field theories, cf. Eq. (34). The system interactions per mode of the ML 4-phasor model scale like $O(N^2)$ and, hence, one would expect a mean-field-like behavior. The outcome is to be compared to the universality class of the REM [67] and of the random homogeneously diluted 4-phasor model [39], displaying $2\beta + \gamma \simeq 2$.

The models we are dealing with suffer of very strong finite size effects. Indeed, the system, as any spin-glass-like system, is known to be Non-deterministic Polynomial Complete (NPC) [72], i.e., operatively, to look for equilibrium states, it occurs a simulation time that scales with the number of modes $N$ approximately as $e^{AN}$. Such equilibration time has been sensitively reduced using the Exchange Monte Carlo algorithm and parallelizing the computation of the energy difference of each single proposed spin update on parallel kernels on GPUs, yet the system stays NPC.

The computation of the energy difference between configurations of complex continuous spins is an essential feature. This particular random laser model displays a connectivity per node growing like $N^2$. Each time a Monte Carlo update is proposed the $\Delta E$ to be computed includes $\mathcal{O}(N^2)$ terms, and likewise increases the time of a single spin update. To cope with this bottleneck in our code each energy contribution is computed apart in parallel on GPU, decreasing the single update to $\mathcal{O}(\ln N)$.

Though much shortened the single mode update time still increases with the size.

Furthermore, our spins are continuous (complex) variables and we have no cunning shortcuts as, for instance, the multi-spin coding that one can exploit for Ising spins, accelerating the computation with bitwise operations and allowing to simulated systems of larger sizes in reasonable computing times.

As a last source of finite size effects, each mode, besides a dynamic phasor value, also has a (quenched) frequency and these influence the connectivity of the mode according to the frequency matching condition (17). Because of this condition modes near the boundaries of the mode spectrum ($k \gtrsim 1$, $k \lesssim N$) interact much less than modes whose frequency lays in the middle of the spectrum ($k \sim N/2$). Though their dynamic evolution and their contribution to the dynamic update is computed in the lasing regime they are less and less important as the external pumping increases. To circumvent this problem in this work we have introduced a slightly different model network, imposing periodic boundary conditions on the frequencies, cf. Eq. (19). This corresponds to work with modes in the central part of the spectrum, as if pertaining to a larger system. Indeed, periodic boundaries turn out to improve the finite size scaling, as if we were working at an effective larger size because of smaller amount of coupling dilution. In this case, the same analysis performed with periodic boundary conditions on the frequencies, with similar sizes and statistics of disordered samples leads to an estimate of $\nu_{eff} = 2\beta + \gamma = 1.2(2)$, that is compatible with a mean-field theory according to the condition (34).

Finally, we have analyzed the low temperature (high pumping) phase of the model with mode networks built on both free and periodic conditions on the frequencies. We find clear evidence for the occurrence of a Replica Symmetry Breaking phase at low temperature. Studying the deviation of the overlap distributions from a Gaussian distribution by standard methods (e.g., the Binder cumulant), as performed in Ref. [39], the onset of such a spin-glass phase can be shown to occur at a temperature consistent with the laser threshold identified by FSS analysis of the specific heat peaks. Introducing PBC also helps in this case as, at the same simulated sizes, the glassy nature of the low $T$ phase is more evident in the model with PBC network, rather than in the one with FBC. This is graphically exemplified in the $P(q)$ shown in Figs. 10, 11.

## VII. ACKNOWLEDGEMENTS

We thank Daniele Ancora and Lorenzo Pinto for useful interaction. We acknowledge the support from the European Research Council (ERC) under the European Union Horizon 2020 Research and Innovation Program, Project LoTGlasSy (Grant Agreement No. 694925). We also acknowledge the support of LazioInnova - Regione

| $N$ | $N_4$ | $T_{\min}$ | $T_{\max}$ | $N_{\mathrm{PT}}$ | $N_{\mathrm{MCS}}$ | $N_{\mathrm{rep}}$ | $N_{\mathrm{s}}$ |
|-----|-------|------------|------------|-------------------|--------------------|--------------------|------------------|
| 18 | $2^8$ | 0.35 | 1.6 | 50 | $2^{19}$ | 4 | 400 |
| 32 | $2^{11}$ | 0.45 | 1.6 | 46 | $2^{19}$ | 4 | 350 |
| 48 | $2^{13}$ | 0.5 | 1.6 | 44 | $2^{20}$ | 4 | 300 |
| 62 | $2^{14}$ | 0.55 | 1.6 | 42 | $2^{20}$ | 4 | 250 |
| 62 | $2^{14}$ | 0.3 | 1.6 | 52 | $2^{20}$ | 4 | 100 |
| 96 | $2^{16}$ | 0.65 | 1.6 | 38 | $2^{20}$ | 2 | 100 |
| 120 | $2^{17}$ | 0.7 | 1.6 | 36 | $2^{20}$ | 2 | 75 |

TABLE II. Details for the simulations of the ML 4-phasor with FBC. Notice that for the size $N = 62$ we performed a second series of simulation with lower $T_{\min}$ and $T_{\max}$, in order to better explore the low temperature phase.

| $N$ | $N_4$ | $T_{\min}$ | $T_{\max}$ | $N_{\mathrm{PT}}$ | $N_{\mathrm{MCS}}$ | $N_{\mathrm{rep}}$ | $N_{\mathrm{s}}$ |
|-----|-------|------------|------------|-------------------|--------------------|--------------------|------------------|
| 18 | $2^9$ | 0.05 | 1.2 | 46 | $2^{19}$ | 2 | 200 |
| 28 | $2^{11}$ | 0.1 | 1.2 | 44 | $2^{19}$ | 2 | 200 |
| 42 | $2^{13}$ | 0.2 | 1.2 | 40 | $2^{20}$ | 2 | 150 |
| 54 | $2^{14}$ | 0.25 | 1.25 | 40 | $2^{20}$ | 4 | 100 |
| 66 | $2^{15}$ | 0.25 | 1.25 | 40 | $2^{20}$ | 2 | 100 |
| 82 | $2^{16}$ | 0.3 | 1.3 | 40 | $2^{20}$ | 2 | 100 |
| 104 | $2^{17}$ | 0.35 | 1.3 | 38 | $2^{21}$ | 2 | 80 |

TABLE III. Details for the simulations of the ML 4-phasor model with PBC.

Lazio under the program Gruppi di ricerca 2020 - POR FESR Lazio 2014-2020, Project NanoProbe (Application code A0375-2020- 36761).

## Appendix A: Numerical Algorithm

The numerical simulations have been performed by means of an Exchange Monte Carlo algorithm [73] parallelized on GPUs to sample the probability distribution Eq. (15). The Exchange Monte Carlo, else called Parallel Tempering (PT) is a very powerful tool for simulating "hardly-relaxing" systems, characterized by a rugged free energy. It is based on the idea that the thermalization is facilitated by a reversible Markovian dynamics of configurations among heat baths at nearby temperatures. In particular, configurations belonging to copies of the system at higher temperature help the copies at lower temperature to jump out of minima of the rugged free energy landscape. For each size $N$ of the simulated systems, we have run PT simulations with $N_{\mathrm{PT}}$ thermal baths at temperature $T_i \in [T_{\min}, T_{\max}]$. The values are reported in Tables II, III.

Each copy at each temperature shares the same realization of quenched disordered couplings $\{J_{\boldsymbol{k}}\}$. Metropolis dynamics is carried out in parallel in each thermal bath and once each 64 steps an exchange (swap) of configurations between baths at neighbouring temperatures is proposed. A swap is proposed sequentially for all pairs of neighbouring inverse temperatures $\beta_i$ and $\beta_{i+1}$, with the following acceptance probability implementing detailed balance with the equilibrium Boltzmann distribution for each thermal bath:

$$p_{\mathrm{swap}} = \min \left[1 \ , \ e^{(\beta_i - \beta_{i+1})(\mathcal{H}[\boldsymbol{a}_i] - \mathcal{H}[\boldsymbol{a}_{i+1}])}\right]. \quad \text{(A1)}$$

For all simulations the $N_{\mathrm{PT}}$ temperatures have been taken with a linear spacing in $T$, that is $T_{i+1} = T_i + \Delta T$, with $\Delta T = 0.025$. On GPUs each thermal bath is simulated in parallel in between swaps.

A further parallelization in the code concerns the bottleneck of the dynamics in dense networks, such as those constructed according to the procedure reported in Sec. III. To compute the energy variation $\Delta E = \mathcal{H}[\boldsymbol{a}'] - \mathcal{H}[\boldsymbol{a}]$, after a spin update $a_k \to a'_k$ has been proposed, requires the sum of $O(N^2)$ terms. The computation of $\Delta E$ has, therefore, been split term by term on parallel kernels on GPU and further resummed in parallalel using $O(\log N)$ operations.

The single update for spins that are complex and spherical is constructed by selecting two spins at random and proposing a random update of both spins which locally preserves their contribution to the global spherical constraint (11). This amounts to extract three pseudo-random numbers for each update proposal.

The code, written in CUDA, has been running on three types of Graphic Processing Units (GPU): Nvidia GTX680 (1536 cores), Nvidia Tesla K20 (2496 cores) and Nvidia Tesla V100 (5120 cores).

The dense ML interaction graph is generated as follows. First, a virtual complete graph with $\binom{N}{4}$ interactions is generated with ordered quadruplets of indices $k_1 < k_2 < k_3 < k_4$. Then the FMC filter is applied to the complete graph either with free or with periodic boundary conditions. We notice that for each ordered quadruplet Eq. (17) can be satisfied only in the permutation $|k_1 - k_2 + k_4 - k_3| = 0$, and any of the other 7 permutations equivalent to it. Each time a quadruplet of indices matches Eq. (17), the corresponding interaction is added to the real graph and a random value extracted from the Gaussian distribution (12) with $p = 4$ is assigned to it. This procedure is repeated picking a quadruplet from the complete graph until a preassigned number $N_4$ of interactions for the ML graph is reached. In order to be able to perform a neat Finite-Size Scaling (FSS) analysis, this number is chosen to be the largest power of 2 below the total number of couplings satisfying the FMC. Each one of the $N_{\mathrm{s}}$ disordered samples simulated is characterized by a realization of the couplings $\{J_{\boldsymbol{k}}\}$ that differs from the others both for the the quadruplet networks and for the numerical value.

To test thermalization we look at energy relaxation on sequential time windows whose length is double each time with respect to the previous one. As a further test we check the symmetry of the overlap distribution $P_J(q)$ - the order parameter of the glass transition, - for single disordered samples. Once dynamical thermalization to equilibrium has been tested and a thermalization time $\tau_{\mathrm{eq}}$ identified, the time average coincides with the canonical ensemble average.

In order to properly estimate statistical errors, time correlations have been taken into account. A correlation time $\tau_{\mathrm{corr}}$ has been identified as the maximum among all thermal bath dynamics, approximately equal to $256 = 2^8$ Monte Carlo steps. Consequently we measure the observables every $\tau_{\mathrm{corr}}$ Monte Carlo steps. If $N_{\mathrm{MCS}}$ is the total amount of Monte Carlo steps of the simulation, for each disordered sample we, thus, have

$$\mathcal{N} \equiv \frac{N_{\mathrm{MCS}} - \tau_{\mathrm{eq}}}{\tau_{\mathrm{corr}}}$$

thermalized, uncorrelated configurations $\boldsymbol{a}_t$ and the ensemble average unbiased estimate is

$$\langle O[\boldsymbol{a}] \rangle = \frac{1}{\mathcal{N}} \sum_{t=\tau_{\mathrm{eq}}/\tau_{\mathrm{corr}}}^{N_{\mathrm{MCS}}/\tau_{\mathrm{corr}}} O[\boldsymbol{a}_t]. \tag{A2}$$

On top of that, we perform simulations on $N_s$ different disordered coupling network samples. That is, for each $\{J_{\boldsymbol{k}}\}$ realization we have a thermal average $\langle O[\boldsymbol{a}] \rangle_J$. Averaging over the random samples yields the least fluctuating finite $N$ proxy for the average in the thermodynamic limit:

$$\overline{O} = \frac{1}{N_s} \sum_{j=1}^{N_s} \langle O[\boldsymbol{a}] \rangle_J^{(j)}. \tag{A3}$$

The statistical error on the average over disorder is much larger that the error on the thermal average and leads to the error bars on the observables displayed in the main text. We observed that taking data uncorrelated in time, in view of the fact that the time average contribution to the error is negligible with respect to the quenched disorder contribution and using the numbers $N_s$ of simulated random samples indicated in table IV, the leading digit of the statistical error practically does not change when it is computed using antidistorsion techniques such as jackknife and bootstrap with respect to a simple standard deviation computation on the sample-to-sample fluctuations. The errorbars in the figures are, therefore, all computed as standard deviations.

## Appendix B: REM numerical study

In the REM, one considers $M = 2^N$ random energy levels as if pertaining to a generic model with $N$ Ising spin variables. The energies $\{E_\nu\}_{\nu \in \{1,...,M\}}$ are extracted as independent Gaussian variables from the distribution function

$$p(E) = \frac{1}{\sqrt{\pi N J^2}} \exp\left(-\frac{E^2}{N J^2}\right), \tag{B1}$$

where the scaling of the variance with $N$ ensures the extensivity of the thermodynamic potentials and $J$ is a

| $N$ | $T_c$ | $\Delta T_c$ | $N_s$ |
|-----|-------|--------------|-------|
| 16 | 0.85 | 0.02 | 300 |
| 20 | 0.83 | 0.02 | 200 |
| 24 | 0.81 | 0.02 | 200 |
| 28 | 0.80 | 0.02 | 100 |

TABLE IV. Details for the simulations of the ML 4-phasor model with PBC.

parameter. An instance of the quenched disorder corresponds to an extraction of the $M$ energy levels. The partition function of the model reads

$$\mathcal{Z} = \sum_{\nu=1}^{M} e^{-\beta E_\nu}. \tag{B2}$$

Data displayed in Fig. 6 were collected through a simple enumeration algorithm, which is described in the following. For each disorder sample of a given system size $N$ the energy levels $\{E_\nu\}$ are generated, by independently extracting a set of $2^N$ random numbers from the Gaussian distribution Eq. (B1) with $J = 1$. A set of equispaced temperatures $T$ is generated in the interval $[T_{\min}, T_{\max}]$, with $T_{\min} = 0.15$ and $T_{\max} = 1.8$ for all sizes. The internal energy of the model is computed as a function of temperature by evaluating the thermal average

$$\langle E \rangle = \frac{\sum_\nu E_\nu \, e^{-\beta E_\nu}}{\sum_\nu e^{-\beta E_\nu}}, \tag{B3}$$

for each of the $\beta = 1/T$ values extracted before. The specific heat is computed from the fluctuations of the internal energy as

$$c_{V_N, i}(T) = \frac{1}{N} \frac{\langle E^2 \rangle - \langle E \rangle^2}{T^2}, \tag{B4}$$

where the index $i$ accounts for the sample. The procedure is repeated for several independent extractions of the random energies $\{E_\nu\}$. Eventually, when a sufficiently large number of samples $N_s$ is collected for each simulated size, the disorder average of the specific heat is computed by averaging over the samples:

$$c_{V_N}(T) = \frac{1}{N_s} \sum_{i=1}^{N_s} c_{V_N, i}(T), \tag{B5}$$

which is the same of Eq. (A3). The error bars in Fig. 6 are given by the statistical error on this average. The number of samples $N_s$ is chosen in such a way that the estimated value of the specific heat remains stable upon fluctuations of the samples included in the average. More details are presented in Table IV.

The finite-size scaling analysis leading to the values of the exponents $\alpha$ and $\nu_{\mathrm{eff}}$ is performed by following the same technique described in App. C.

## Appendix C: Finite-size scaling details

In order to assess the critical temperatures $T_c(N)$ in Table I we fit the points around the peak of each curve in Figs. 7, 8 with a quadratic function of the temperature $f_N(T) = a_N + b_N T + c_N T^2$ and compute the maximum point of each fitting function as $T_c(N) = -b_N/(2c_N)$, estimating the statistical error accordingly. The critical temperature $T_c(\infty)$ of the model can be extrapolated from the fit of the finite-size critical temperatures with the following function: $T_c(N) = T_c(\infty) + aN^{-b}$, where the exponent $b$ gives a first rough estimate of the critical exponent $1/\nu_{\text{eff}}$. The results of the fit are:

$$\text{FBC: } \quad T_c(\infty) = 0.86 \pm 0.03, \quad b = 1.6 \pm 0.5, \quad \text{(C1)}$$

$$\text{PBC: } \quad T_c(\infty) = 0.61 \pm 0.03, \quad b = 0.98 \pm 0.3. \quad \text{(C2)}$$

We then take the following Ansatz on the form of scaling function $\hat{f}$ in Eq. (38)

$$\hat{f}(x) = A + Cx^2, \quad \text{(C3)}$$

where $x = N^{1/\nu_{\text{eff}}} t_N$, with $t_N$ computed by using the $T_c(N)$ reported in Table I. In the previous Ansatz we have not included the linear term, since the points are translated in order for the peak of each curve to be in the origin and we expect the linear term not to matter. With this Ansatz the scaling hypothesis for the specific heat Eq. (38) reads as

$$c_{V_N}(T) = \tilde{A}_N + \tilde{C}_N t_N^2 \quad \text{(C4)}$$

where $\tilde{X}_N = X_N N^{\frac{\alpha+m}{\nu_{\text{eff}}}}$, with $X_N = \{A_N, C_N\}$ and $m = \{0, 2\}$. We fit the points of the curves around the critical temperature with the previous function and determine the values of the coefficients. We, then, notice that the behaviour of the logarithm of the coefficients,

$$\ln \tilde{A}_N = \ln A_N + \frac{\alpha}{\nu_{\text{eff}}} \ln N,$$

$$\ln |\tilde{C}_N| = \ln |C_N| + \frac{\alpha+2}{\nu_{\text{eff}}} \ln N$$

is linear in $\ln N$ and the estimates of $\alpha$ and $\nu_{\text{eff}}$ can be obtained by linear interpolation.

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

(1981).