# Peer review of "Universality class of the mode-locked glassy random laser"

_SciPost Physics_

## Round 1 · Referee Report · Carlo Vanoni (Referee 1) · 2022-12-27

Strengths

1- The authors present solid numerical results for the mode-locked 4-phasor model and effectively combine them with analytical predictions based on mean-field scaling arguments.

2- The model and the motivations for studying it are extensively discussed in the Introduction, which is very well-written and gives a complete presentation of the subject.

3- The authors exploit modified boundary conditions (PBC instead of FBC) for the frequencies to reduce the finite-size effects dramatically and obtain more accurate predictions of critical exponents.

Report

The authors present strong evidence that the critical properties of the mode-locked 4-phasor model, characterized by the Frequency Matching Condition (FMC), can be described by a mean-field theory. This is achieved by combining extensive numerical simulations, analytical arguments based on the scaling hypothesis to control the finite-size effects, and the introduction of periodic boundary conditions on the frequencies to reduce finite-size effects in the numerics.
The problem addressed in the paper is notoriously very difficult and relevant both from a theoretical and experimental point of view. The techniques and results discussed represent a step forward in the comprehension of the model considered.
On this basis, I think the paper fully meets the Journal's criteria. I, therefore, suggest publication once some minor issues will be addressed by the authors.

Requested changes

1- Just above Eq. (10), the authors introduce the ratio $\mathcal{P}$ between the pump rate and the spontaneous emission rate. According to what the authors say, the spontaneous emission rate is proportional to the temperature of the bath $T$, while the pump rate is proportional to the energy of the system $\mathcal{E}=\epsilon N$. Consequently, I would expect $\mathcal{P} \propto \epsilon/T$. However, Eq. (10) states that $1/\mathcal{P}^2 = T/\epsilon^2$, which seems to be inconsistent with the definition. Can the authors clarify this point?

2- In Sec. V the authors present numerical results for the overlap distribution across the glass transition. They say they used $\mathcal{N}$ equilibrium configurations of replicas to accumulate statistics, but only in Appendix A the authors relate the quantity $\mathcal{N}$ to the Monte Carlo steps of the algorithm and no reference to the equation in the Appendix is present in the main text. I would suggest adding a reference to the Appendix so that the reader can have a better understanding of the meaning of $\mathcal{N}$. Moreover, neither in the caption of Fig. (9) nor of Fig. (10) there is an indication of the typical value of $\mathcal{N}$ in the numerical results presented. Is it possible to add it?

3- In Eq. (2) the notation $\sum_{\kappa|\mathrm{FMC}(\kappa)}$ is not immediately clear as there is no reference to Eq. (4), where the meaning of $\mathrm{FMC}(\kappa)$ is presented. I suggest adding a reference to Eq. (4) below Eq. (2) to ease the reading.

4- In order to facilitate the understanding of the plots, I would add a "$T$" beside the color maps in Figs. (1), (3), (4), (5), (10), and (11), so that it is immediately clear what different colors stand for.

5- There are some typos throughout the paper. In the abstract "using , with" doesn't seem to be correct. At the beginning of page 2, in the sentence starting with "In standard mode-locked lasers ..." there is a parenthesis that is opened but not closed. Just before Eq. (5) "dymamics" should be replaced with "dynamics".

---

## Round 1 · Referee Report · Benjamin Guiselin (Referee 2) · 2023-1-11

Strengths

1- The authors have performed extensive computer simulations of the Mode-Locked 4-phasor model beyond the narrow-bandwidth approximation.

2- The authors have implemented a delicate finite size scaling (FSS) analysis, and the procedure is clearly explained.

3- The authors have introduced a new set of boundary conditions for this problem, to ease their FSS analysis.

Weaknesses

1- The justification of the model used by the authors is not totally well explained.

2- The arguments sustaining the mean-field scaling laws proposed by the authors are not elaborated enough.

Report

The paper of Niedda et coworkers studies the Mode-Locked 4-phasor model beyond the narrow-bandwidth approximation, for which mean-field theory cannot be computed exactly, because the model is no longer fully-connected. They thus resort to extensive simulations on GPUs in order to address the statistical properties of the model. In particular, they show that its thermodynamic properties are similar to the ones of mean-field glassy models, for which a one-step replica symmetry breaking occurs at a static transition temperature.

In order to demonstrate this important result, the authors introduce new boundary conditions to have a better control on finite-size effects. This allows them to perform an extensive finite size scaling analysis of the heat capacity of the model for various system sizes, from which critical exponents can be extracted to compare with mean-field theory. They eventually discuss the replica symmetry breaking at the lasing threshold and determine the Parisi order parameter for glassy systems, namely, the overlap probability distribution among samples for the same realization of the disorder (which is eventually averaged over the disorder).

The paper is rich and overall well-written. However, I found hard to understand some of the arguments regarding the scaling laws. As a result, I recommend publication in SciPost Physics if the following points are taken into account by the authors.

  • As previously said, the main problem I encountered while refereeing is about the scaling laws provided by the authors. They claim that the critical exponents for a standard second order critical point (and also for the REM) are $\nu=2$ and $\alpha=2/3$. However, I naively thought that they should be $\nu=1/2$ and $\alpha=0$ (as for standard scalar $\phi^4$-theory). To justify their claim, the authors use a scaling argument that I find hard to understand. In particular, they approximate the fluctuations below the critical point by the distance of the minimum of the Landau free energy from the origin, while the curvature of the Landau free energy at the minimum should be considered instead.

  • I do not understand the last paragraph of the left column page 3, where it is written that the Fourier transform of $a_k$ should be roughly a Dirac delta function at $\omega_k$. Indeed, $a_k$ should be a slowly-varying amplitude, and thus its Fourier transform should only display low frequencies centered around 0, and not around $\omega_k$. I guess the authors discuss the Fourier transform of the electric field rather than the one of the slow amplitude $a_k(t)$, but maybe I am missing a point here.

  • Second paragraph of page 4, I do not understand the end of the sentence ", and in the case of cavityless systems also compensate the leakages". I suspect one word is missing or replaced by another one.

  • Just above Section III, the authors say that "As already mentioned (and without loss of generality) we will consider the J's as real parameters". I could not find to which part of the paper the authors were referring. Does it mean that the authors assume to be in the purely dissipative regime?

  • At the beginning of the third paragraph page 5, the authors simplify the problem in the limit $\gamma\ll \delta\omega$, and conclude that $\mathcal{H}_2$ is an additive constant to the Hamiltonian. I do not understand why, since the couplings are a priori different from one mode to another, and the spherical constraint cannot be factorised out.

  • Is there a reason why Eq. (19) is not just written $|k_a-k_b| = N - |k_a-k_b| if |k_a-k_b| \gtrsim N/2$? Besides $N/2$ is not well defined for odd $N$, and so the integer part should be considered.

  • In Appendix B, I do not understand the argument for the choice of the scaling function [below Eq. (B3)]. Could you explain more?

  • At the end of Section IV, the authors conclude that the model belongs to a mean-field universality class because the value of $1/\nu$ is consistent with this statement. Actually, the only rigorous statement that can be made is that the universality class of the model is not incompatible with a mean-field one. How do the authors conclude that the universality class is indeed a mean-field one? I think I am missing a point here.

  • Could you justify why the complexity for one step goes from $N^2$ to $\ln N$ via parallelisation on GPUs, as claimed in the Conclusion?

  • In the Conclusion, the authors claim that the RSB occurs at the lasing threshold. But from the data of Fig. 11, it seems that the the secondary peaks in the overlap distribution appear at much lower temperature (about 0.4) than the critical temperature $T_c$ (about 0.6). Is it expected that the two temperatures coincide in the large $N$ limit?

In addition, I noticed few typos in addition to the ones already noted by the other referee, that I list below:

  • Last paragraph of page 2, there is a typo in "heterogeneous".

  • Last paragraph of page 3, there is a typo, it should be "responsible for the Kerr effect".

  • Beginning of the second column page 4, there is a typo in "heterogeneous".

  • Just after Eq. (2), $\sigma_p^2\sim N^{p-2}$ should be replaced by $\sigma_p^2\sim N^{2-p}$.

  • There is a problem of reference in the footnote [61].

  • At the end of the first column page 8, there is an "in" missing between "presented" and "following subsection".

  • One should change the notation for the exponent $x$ below Eq. (B4) since $x$ is already used for something else above.

  • If I understand correctly, the coefficient $C$ in Eq. (B3) should be negative (so that the heat capacity is maximum at the transition). One should thus add absolute values in the very last equation of Appendix B because $\ln C_N$ and $\ln \tilde C_N$ are not properly defined.

  • In the caption of Fig. 9, "differ from each other" should be "differs from the others".

Requested changes

1- Elaborate the scaling arguments.

2- Clearly indicate the assumptions behind the model used eventually for the simulations (only a quartic coupling, and real parameters).

3- Provide the simulation details for the REM (at least in an Appendix).

4- Discuss where the error bars are coming from in Figs. 6, 7 and 8 (at least in the Appendix).

5- Clarify the Appendix B about the finite size scaling analysis protocol.

6- Clarify the argument to conclude that the model is of mean-field nature.

7- Correct the various typos.

---

## Round 2 · Referee Report · Carlo Vanoni (Referee 1) · 2023-2-1

Report

In the second version of the manuscript, the authors have clarified the doubts I had about some points of their manuscript; in particular the points (1) -(4) in my report.
I have noticed, however, that the typos identified at point (5) of my report have not been addressed, so I wonder if this is because the authors believe they are not typos or if it is just a forgetfulness. I report here the typos I identified

- In the abstract "using , with" doesn't seem to be correct.
- At the beginning of page 2, in the sentence starting with "In standard mode-locked lasers ..." there is a parenthesis that is opened but not closed.
- Just before Eq. (5) "dymamics" should be replaced with "dynamics".

In any case, I think the manuscript, in its present form, is clear and can proceed for publication.

  • validity: -
  • significance: -
  • originality: -
  • clarity: -
  • formatting: -
  • grammar: -

Author:  Luca Leuzzi  on 2023-02-24  [id 3403]

(in reply to Report 1 by Carlo Vanoni on 2023-02-01)

We apologize for our forgetfulness. We have corrected all the typos indicated by the referee in the revised version.

Best regards
Luca Leuzzi, on behalf of all the authors.

---

## Round 2 · Referee Report · Benjamin Guiselin (Referee 2) · 2023-2-10

Report

The authors have taken into account most of the comments I made about the first version of their work. This second version is far clearer. I have though still few minor remarks:

- To be consistent with the comment from my previous report, the sentence "The outcome of the scaling analysis is that the mode-locked random laser is in a mean-field universality class" in the abstract should be toned down, as the work of the authors only show that the data are compatible with a mean-field universality class.

- In page 2, sparse networks are not introduced. At least, the scaling of the number of interactions wit $N$ should be indicated.

- In Eq. (36), $\xi_\infty$ should be to power $d$ so that one recovers $\omega=\psi/\nu$.

- What $\mathbf{A}$ stands for in Eqs. (A2) and (A3)?

- The authors did not fully take into account my comment about the computation of error bars. In Appendix A, they mention that the error bars come from the disorder average, but I am wondering how they compute them in practice? Is it the standard deviation of the canonical averages for several realizations of the disorder, or is it a more sophisticated quantity (for instance obtained via the bootstrap or the jackknife method)?

Several typos should also be corrected:

- In the abstract "using , with" looks like a mistake.

- Third line of the right column page 2: "do not allow" should be "does not allow".

- Second paragraph of the right column page 2: "significantly differ from the homogeneous mean-field" should be "significantly differs from the homogeneous mean-field".

- Third line of the last paragraph in the right columm page 2: "etherogeneous" should be "heterogeneous".

- Second line of the fourth paragraph in the left column page 3: "reads as" should be "reads".

- Fourth line of the second paragraph in the right column page 3: "dymamics" should be "dynamics".

- Second paragraph of the left column page 4 the sentence: "and in the case of cavityless systems also compensate the leakages" has not been corrected by the authors as they claimed in the response to my comment. This correction should be done.

- Third line of the second paragraph in the right columm page 4: "etherogeneous" should be "heterogeneous".

- Seventh line of the second paragraph in the right columm page 4: "reasons" should be "reason".

- Third line before the end of the left column page 8: "is the scaling the" should be "is the scaling of".

- Last line of the before last paragraph in the right column page 8: "estimate of the exponent" should be "estimate of the exponents".

- First line of the left column page 12: '$\mathcal{O}(N^2)$' should be $O(N^2)$.

- Last line of the second paragraph in the left column page 12: "GPU's" should be "GPUs".

- Third paragraph in the left column page 12: all '$\mathcal{O}$' should be replaced by '$O$'.

- Third line of the first paragraph in the left column page 13: "GPU's" should be "GPUs".

- First line of the second paragraph in the left column page 13: "share" should be "shares".

- First line of the first paragraph in the right column page 13: "GPU's" should be "GPUs".

Once these remarks and typos are taken into account by the authors, I naturally support publication of this manuscript in SciPost Physics.

  • validity: top
  • significance: top
  • originality: high
  • clarity: good
  • formatting: excellent
  • grammar: acceptable

Author:  Luca Leuzzi  on 2023-02-24  [id 3402]

(in reply to Report 2 by Benjamin Guiselin on 2023-02-10)

In the following we answer point to point to the referee remark.

Best regards Luca Leuzzi, on behalf of all the authors.

Report 2

Referee: The authors have taken into account most of the comments I made about the first version of their work. This second version is far clearer. I have though still few minor remarks: To be consistent with the comment from my previous report, the sentence "The outcome of the scaling analysis is that the mode-locked random laser is in a mean-field universality class" in the abstract should be toned down, as the work of the authors only show that the data are compatible with a mean-field universality class.

Reply: we changed the text in the abstract as “The outcome of the scaling analysis is that the mode-locked random laser universality class is compatible with a mean-field one”.

Referee: In page 2, sparse networks are not introduced. At least, the scaling of the number of interactions with N should be indicated.

Reply: We added a footnote, [46], where we explain that “By sparse networks we mean that the average connectivity of each variable does not scale with the number $N$ of variable and, therefore, the total number of couplings in the systems grows like $N$.”

Referee: - In Eq. (36), ξ∞ should be to power d so that one recovers ω=ψ/ν

Reply: The referee is right, we corrected the formula as observed.

Referee: What A stands for in Eqs. (A2) and (A3)?

Reply: It should have been $a$, rather than $A$, and it is a mode configuration. We changed the lettercase and we added the symbol $\bm a_t$ after configurations.

Referee: The authors did not fully take into account my comment about the computation of error bars. In Appendix A, they mention that the error bars come from the disorder average, but I am wondering how they compute them in practice? Is it the standard deviation of the canonical averages for several realizations of the disorder, or is it a more sophisticated quantity (for instance obtained via the bootstrap or the jackknife method)?

Reply: We are sorry for that. Using jackknife (or bootstrap) did not change a lot the outcome, possibly because we can basically neglect the thermal fluctuations with respect to the quenched disorder fluctuations in computing the statistical error. We explain this in Appendix A in the revised version: “We observed that taking data uncorrelated in time, in view of the fact that the time average contribution to the error is negligible with respect to the quenched disorder contribution and using the numbers $N_s$ of simulated random samples indicated in table \ref{tab4}, the leading digit of the statistical error practically does not change when it is computed using anti-distorsion techniques such as jackknife and bootstrap with respect to a simple standard deviation computation on the sample-to-sample fluctuations. The errorbars in the figures are, therefore, all computed as standard deviations.

Referee: Several typos should also be corrected

Reply: We thank the referee for the careful reading. We have corrected all the typos indicated.

---

## Round 2 · Author Response

We thank both referees for a thourough reading of our manuscript and theor comments and observations. They certainly helped improving the presentation of our work. In the following we reply in detail to each comment of the referees.

Report 2 REFEREE: As previously said, the main problem I encountered while refereeing is about the scaling laws provided by the authors. They claim that the critical exponents for a standard second order critical point (and also for the REM) are ν = 2 and α = 2/3. However, I naively thought that they should be ν = 1/2 and α = 0 (as for standard scalar φ4 -theory). To justify their claim, the authors use a scaling argument that I find hard to understand. In particular, they approximate the fuctuations below the critical point by the distance of the minimum of the Landau free energy from the origin, while the curvature of the Landau free energy at the minimum should be considered instead. REPLY: The referee is absolutely right and we apologize for being sloppy in the presentation. We unproperly used the symbol $\nu$ for $\nu\times d$, $d$ being the dimension of the underlying short-range interacting lattice. Indeed, if the theory if a $\phi^4$ theory, in the limit of validity of the mean-field theory it is $\nu = 1/2$, and $\nu d$ = 2 because the upper ctitical dimension is 4. Furhermore, we were also unclear in distinguishing the theoretical mean-field value of the specific heat exponent $\alpha=0$ 0 from the numerical estimates obtained from finite size scaling, as, e.g., $\alpha\simeq 2/3$ in a previous study of the ML random laser [Ref 20], or $\alpha\simeq 0.52$ in the numerical simulations of the REM (who is known to be in the universality class of the \phi^4 mean-field theory). The scaling argument, actually, directly concerns the susceptibility and the order parameter behavior around criticality, that is, a relationship for the β and γ exponents: 2 β +γ = 2 (in general 2 β +γ = n/(n-2) for a φ^n theory). Because of the hyperscaling relation 2 β +γ = ν d, in a φ^4 theory at the upper critical dimension this corresponds to what we errenously termed ν (and also referred to as “the correlation length critical exponent” in at least one occasion) in the old version of the paper. We have been rewriting the section 4, and part of the conclusions, making explicit the role of β and γ exponents in the scaling argument. We introduce the term ν_{eff} to sometimes shorten 2 β +γ. In the text of the revised version we explicitly report that the α exponent (estimated from the finite size peak of the specific heat) suffers of strong pre-asymptotic effects, as far as the simulated sizes are small. This can be observed already in the simpler mean-field cases, as the REM, where the estimated 2β +γ ~ 2 is compatible with 2 but the estimate for $\alpha$ is non zero. This finite size pre-asymptotic effect for α is detected also in the mode-locked $4$-phasor model.

— REFEREE: I do not understand the last paragraph of the left column page 3, where it is written that the Fourier transform of ak should be roughly a Dirac delta function at ωk . Indeed, ak should be a slowly-varying amplitude, and thus its Fourier transform should only display low frequencies centered around 0, and not around ωk . I guess the authors discuss the Fourier transform of the electric field rather than the one of the slow amplitude ak(t), but maybe I am missing a point here. The referee is right also here. There is a mistake in the formula. When we perform the Fourier transform of the em field (3) on fast time scales (inverse of the frequencies of the normal modes waves oscillations), i.e., on a time window $2\tau$ large enough so that many oscillations take place ($1/\omega_k \ll \tau)$ we have
 \sum_k \int_{t-\tau}^{t+\tau} ds/2\pi e^{-\imath ω s} a_k(s) e^{\imath ω_k s} 
that in the slow amplitude approximation (a_k(t) does not change on time scale $\tau$) is approximately equal to 
 \sum_k a_k(t) \int_{t-\tau}^{t+\tau} ds /2\pi e^{-\imath (ω-ω_k) s} \simeq
 \sum_k a_k(t) \delta(ω-ω_k).
We are left with a slowly varying amplitude a_k(t) times a \delta (ω-ω_k). One may write, as well, a_k(t) = a(t,ω_k). We corrected the text accordingly. —
REFEREE: Second paragraph of page 4, I do not understand the end of the sentence ", and in the case of cavityless systems also compensate the leakages". I suspect one word is missing or replaced by another one. REPLY: We thank the referee to pint this out, we added a subject to the sentence:
“and in the case of cavityless systems also compensate the leakages.” becomes
“and in the case of cavityless systems energy pumping also compensate the leakages.” — REFEREE: Just above Section III, the authors say that "As already mentioned (and without loss of generality) we will consider the J's as real parameters". I could not find to which part of the paper the authors were referring. Does it mean that the authors assume to be in the purely dissipative regime? REPLY: Yes, we changed the sentence in “As already mentioned in section II when introducing the dissipative limit we will consider the J's as real parameters, without loss of generality.” —
REFEREE: At the beginning of the third paragraph page 5, the authors simplify the problem in the limit γ << \delta ω , and conclude that H2 is an additive constant to the Hamiltonian. I do not understand why, since the couplings are a priori different from one mode to another, and the spherical constraint cannot be factorised out. REPLY: The Referee is right also in this respect. If (16) holds we have N modes with N distinct frequencies and the H_2 is diagonal: \sum_J_k |a_k|^2. Our statement of additive constant only holds if J(k) is homogeneous in k. The diagonal part of the second order contribution to the Hamiltonian represents the gain profile of the optical random medium. As a working hypothesis we are assuming a uniform gain profile over the whole spectrum. 
We amended the revised text accordingly. —
REFEREE: Is there a reason why Eq. (19) is not just written
|ka - kb| = N - |ka - kb| if |ka - kb| \gtrsim N/2? Besides N/2 is not well defined for odd N , and so the integer part should be considered. REPLY: We agree that it is more immediate as the referee. Wa emended the text accordingly, including [N/2] (integer part of), in the condition (19). —
REFEREE: In Appendix B, I do not understand the argument for the choice of the scaling function [below Eq. (B3)]. Could you explain more? REPLY: The scaling function $\hat f(x)$ retraces the behavior of f_N(T) around T_c. That is, the linear term is not there because the maximum of the parabola is in $t_N=0$, by definition of $t_N=T-T_c(N)$. Both in $f_N$ and in $\hat f$ we might have added a third order term but we have tested that it becomes irrelevant in the critical region. —
REFEREE: At the end of Section IV, the authors conclude that the model belongs to a mean-field universality class because the value of 1/ν is consistent with this statement. Actually, the only rigorous statement that can be made is that the universality class of the model is not incompatible with a mean-field one. How do the authors conclude that the universality class is indeed a mean-field one? I think I am missing a point here. REPLY: We conclude that the universality class is compatible (i.e., non incompatible) with a mean-field universality class, though strong, possibly pre-asymptotic, finite size effects prevent us from saying what universality class. So we basically agree with the referee. Rather than “assess the mean-field nature” or “yield a mean-field universality class”, we have now limited ourselves to say that our best data, obtained with pbc, yield a picture that is compatible with a mean-field class of universality. 
We changed the text accordingly. For instance “we can assess the mean-field nature of the glass transition in the ML 4-phasor model.” becomes “we observe that the glass transition of the ML 4-phasor model is compatible with a mean-field transition”. —
REFEREE: Could you justify why the complexity for one step goes from N 2 to ln N via parallelisation on GPUs, as claimed in the Conclusion? REPLY: At each update proposal of a given couple of modes the new \Delta H is computed, that is all the contributions to the Hamiltonian involving the two modes are changed. These are O(N^2) and involve the product of the quadruplets mode amplitudes times the relative coupling constant.
If we split the computation of each one of these contribution on different kernels on a GPU, they are all computed in parallel, synchronously. If we increase N, the computation of the products yielding each quadruplet contribution to the energy does not scale with N because it is carried out in parallel. We are left with the operation of sum, only. This can be carried out summing - in parallel - couples of quadruplets contributions, then we sum the N^2/2 remaining numbers as couples, than the N^2/4 numbers left are summed in couples and so on until we reach a single number: the total sum. For instance, if N^2=16 it takes 4 steps, log_2 16, to sum the total energy. If N^2=64 it takes 6=\log_2 64 steps. In general it takes \sim \log N^2 = 2 \log N steps to sum O(N^2) terms. —
REFEREE: In the Conclusion, the authors claim that the RSB occurs at the lasing threshold. But from the data of Fig. 11, it seems that the the secondary peaks in the overlap distribution appear at much lower temperature (about 0.4) than the critical temperature Tc (about 0.6). Is it expected that the two temperatures coincide in the large N limit? REPLY: Numerically, at finite $N$, the critical region is around the temperature at which deviations from gaussianity occur in the P(q). Eventually, at low enough T, deep in the spin glass phase, side peaks arise but the transition already occurrs at higher T, when the P(q) at finite N is no longer a Gaussian. This is not easy to be detected by the eye and one has to rely on specific observables such as the Binder cumulant (basically the deviation of the kurtosys from 3). The analysis procedure was extensively presented in Ref [40] so that we did not report it here.
However, we feel that we have to clarify the point raised by the referee and we modified the text in the conclusions as follows: “We find clear evidence for the occurrence of a Replica Symmetry Breaking phase at low temperature. Studying the deviation of the overlap distributions from a Gaussian distribution by standard methods (e.g., the Binder cumulant), as performed in Ref. [40], the onset of such a spin-glass phase can be shown to occur at a temperature consistent with the laser threshold identified by FSS analysis of the specific heat peaks.”

Eventually, on request of the referee, we have 1- Elaborated the scaling arguments. 

2- Indicated the assumptions behind the model used eventually for the simulations (only a quartic coupling, and real parameters).

3- Provided the simulation details for the REM in a new Appendix, B.

4- Discussed in the Appendices A and B where the error bars are coming from in Figs. 6, 7 and 8. 5- Clarified the FSS protocol in the Appendix C (former B). 6- Clarified the argument to conclude that the model is compatible with a mean-field theory.

7- Corrected the various typos.

Report 1

REFEREE
1- Just above Eq. (10), the authors introduce the ratio P between the pump rate and the spontaneous emission rate. According to what the authors say, the spontaneous emission rate is proportional to the temperature of the bath T , while the pump rate is proportional to the energy of the system E = \epsilon N . Consequently, I would expect P \propto \epsilon/T . However, Eq. (10) states that 1/P 2 = T /\epsilon2 , which seems to be inconsistent with the deRnition. Can the authors clarify this point? REPLY 
The sentence the referee is referring to is, indeed, confusing. We have corrected it in the revised version of the paper. In fact, with the notation P it is meant the pumping rate, which is defined as the inverse of the effective ("photonic") temperature squared. 
We modified the text as:
“However, a stationary regime can be described as if the system is at equilibrium with an effective thermal bath, whose effective temperature (a ``photonic'' temperature) accounts both for the amount of energy ${\mathcal{E}}=\epsilon N$ stored into the system because of the external pumping and for the spontaneous emission rate. The latter is proportional to the kinetic energy of the atoms, e.~g., to the heat bath temperature $T$. Eventually, the external parameter driving the lasing transition turns out to be [34,35,50]
 T_{\rm photonic} = \frac{T}{\epsilon^2}.
One can also introduce the pumping rate $\mathcal{P}$ [42] as the inverse of the square root of this ratio:
 \mathcal P^2 = \beta \epsilon^2 = \beta_{\rm photonic} “ REFEREE
2- In Sec. V the authors present numerical results for the overlap distribution across the glass transition. They say they used N equilibrium configurations of replicas to accumulate statistics, but only in Appendix A the authors relate the quantity N to the Monte Carlo steps of the algorithm and no reference to the equation in the Appendix is present in the main text. I would suggest adding a reference to the Appendix so that the reader can have a better understanding of the meaning of . Moreover, neither in the caption of Fig. (9) nor of Fig. (10) there is an indication of the typical value of N in the numerical results presented. Is it possible to add it? REPLY
Certainly we are adding this information and we thank the referee to point this out. The number of configurations $\mathcal N$ actually used from our data can be evinced from the fifth column of tables \ref{tab2}-\ref{tab3} in appendix \ref{app0}, in which the last half of the simulated Monte Carlo steps are surely thermalized and the correlation time was estimated to be $2^8$ Monte Carlo steps. Eventually, for each realization of the quenched random couplings we have $\mathcal N=2^{10}-2^{12}$, depending on the size. We added this information in the text with the reference to Appendix A. Furthermore in appendix A we added the valure of the correlation time (256 Monte Carlo steps).

REFEREE
3- In Eq. (2) the notation \sum_{\kappa | FMC(\kappa)} is not immediately clear as there is no reference to Eq. (4), where the meaning of FMC( ) is presented. I suggest adding a reference to Eq. (4) below Eq. (2) to ease the reading. REPLY
We aniticipated a reference to Eq. (4), and also to (1), soon after Eq. (2). REFEREE
4- In order to facilitate the understanding of the plots, I would add a "T " beside the color maps in Figs. (1), (3), (4), (5), (10), and (11), so that it is immediately clear what different colors stand for. REPLY
We added the label T in all figures and in their captions as well.

---

## Round 2 · List of Changes

Page 3, left column, last paragraph after Eq. (2): “where the expression of the sum over the indices k satisfying a FMC, like (1), will be soon clarified in Eq. (4).” has been added.
Page 3, left column, last paragraph after Eq. (3): “taking the Fourier transform of Eq. (3), “ has been added and “implies that a (t) ≃ a (t, ω)δ(ω − ω ). “ has been corrected.
Page 4, first column, second paragraph: we changed “whose effective temperature (a “photonic” temperature) is related to the ratio P between the pump rate and the spontaneous emission rate. “
in
“whose effective temperature (a “photonic” temperature) accounts both for the amount of energy E = εN stored into the system because of the external pumping and for the spontaneous emission rate. “
Equation 10: we took away 1/P^2 here and after Eq. (10) we added
“One can also introduce the pumping rate P [42] as the inverse of the square root of this ratio: $$P^2=\epsilon^2/T=1/T_{\em photonic}$$.
Page 5, first column, fourth paragraph: we changed “and together with the spherical constraint Eq. (11), it becomes an irrelevant additive constant. “ with 
“If we assume that the diagonal part of the pairwise couplings does not depend on the modes, together with the spherical constraint Eq. (11), this term is an irrelevant additive constant. The diagonal part of the linear contribution to the Hamiltonian physically represents the gain profile of the optical random medium (possibly becoming a ran- dom laser at high pumping). As a working hypothesis we are assuming a uniform gain profile over the whole spectrum. ”
Page 6, Equation (19): we changed the formula as reported in the reply to referee 2
Page 7, Section IV, first paragraph substituted by the following three paragraphs
“In a φ4 mean-field theory (a Landau theory) the critical exponents characterizing the universality class are β = 1/2 for the order parameter ⟨φ⟩, γ = 1 for the sus- ceptibility χ and ν = 1/2 for the correlation length. They satisfy the hyperscaling relation 2β + γ = νd, holding for all dimensions d ≤ duc, the upper critical dimension, that is duc = 4 in a φ4 model. As an instance, this is the universality class of the Random Energy Model (REM), a reference simplified model for the glass transi- tion. This is also the universality class of the mean-field 4-phasor model representing a random laser in the so- called narrow-band approximation, both in a fully connected interaction network, where the solution can be analytically computed [34] and in a uniformly randomly diluted version of the model, analyzed by means of equilibrium Monte Carlo simulations in Ref. [40]. Moving to the more realistic random laser models, where the basic ingredient for mode-locking, the frequency matching condition (1) is implemented, it is more difficult to understand whether the universality class re- mains the same. In Ref. [40] an estimate of the value of the critical exponent νeff ≡ 2β + γ ≃ 2/3 was provided for the mode-locked random laser model. This result is quite different from the value 2β + γ = 2 which charac- terizes the REM model, even if we consider its numerical finite-size scaling analysis. 
As an instance, the REM specific heat behaviour for small sizes N = 16, 20, 24, 28 is reported in Fig. 6. Details about the numerical technique used are given in App. B. Even though the simulated N are not very large, from the interpolation of the cV (T) peaks it turns out that νeff = 2β + γ = 1.9 ± 0.2. Strong finite size effects are there, as one can observe from the estimate of the α exponent, displaying a value α = 0.52 ± 0.07, rather different from the mean-field exponent α = 0. Be- cause of preasymptotic effects, indeed, the scaling relation 2β + γ + α = 2 (independent from the system dimension) appears to be violated. “
Former Eq. (20) has been erased.
In IV.A, page 7, second column, in Eq. (21) (former 22) $\propro \chi…$ has been added.
Page 8, first colunm, after Eq. (29) replacing “This result reproduces the value 1/ν = 1/2 for the scaling exponent of the REM. “ the following paragraph and formula have been added
“Let us recognize that Eq. (27) is the susceptibility, cf. Eq. (21), whereas Eq. (28) is the scaling of the square of the order paramater ⟨φ⟩ = φ∗, which is scaling as φ^* ~|\tau|^β. Therefore 
 |τ| ∼ 1/N1/(2β+γ)
In the mean-field φ4 theory 2β + γ = 2. Since the upper critical dimension is duc = 4 this corresponds to νduc ≡ νeff = 2, i.e., ν = 1/2 for the mean-field critical correlation length exponent. “

Page 8, second column, in Eq. (30), former (31), 2n has been substituted by n, an consistently so in Eqs. (31,32,33). After Eq. (32) the expression 
β = 1/(n-2)
has been added.
Former Eqs. (34, 35) have been modified and substituted by current Eqs. (32,33)
Page 8. The paragraph “This result implies that, in order to be compatible with mean-field theory, the values of 1/ν must fall in an in- terval defined by taking n = 2 and n → ∞ in the previ- ous expression. Eventually, the critical exponent for the scaling of the specific heat width in a generic mean-field theory must take value in the interval 
1/2 ≤ ν < 1. 
The previous argument is exact only in the large-N limit where the saddle-point approximation holds. “
has been changed into
“This result implies that, in order to be compatible with mean-fieldtheory,thevaluesofνeff =2β+γ=n/(n−2) must fall in an interval defined by taking n = 4 and n → ∞ in the previous expression. Eventually, the critical exponent for the scaling of the specific heat width in a generic mean-field theory must take value in the interval 
1≤νeff ≤2.
Given the specific theory φ and its upper critical dimen- sion duc(n), the critical mean-field exponent ν is equal to ν=νeff/duc(n). In the model under consideration, though, we have a dense (though not fully connected) interaction network and we do not have a reference d-dimensional lattice un- derneath, such that a scaling relation of the number of modes to a characteristic length can be set, as, for instance N = Ld in a d-dimensional hypercubic lattice. Our analysis will, therefore, be limited to the estimate of the exponent α, β and γ. It is also worth noting that the previous argument is exact only in the large-N limit, where the saddle-point approximation holds. “

In Eqs. (40,41), in the figures and throughout the text when needed ν is replaced by ν_eff, or, equivalently, by 2 β+ γ.
Page 9, second column, last sentence of Section IV “we can assess the mean-field nature of the glass transition in the ML 4-phasor model. “ is replaced by “we observe that the glass transition of the ML 4-phasor model is compatible with a mean-field transition. ”
Page 10, second column, Section V. We added the sentence “The number of configurations N actually used from our data can be evinced from tables II-III in appendix A, in which the last half of the simulated Monte Carlo steps are surely thermalized and the correlation time was estimated to be 28 Monte Carlo steps. Eventually, for each realization of the quenched random couplings we have N = 210 − 212, depending on the size. “ in the third paragraph.

Page 12, second column, last sentence of the first paragraph has been modified in “In this case, the same analysis performed with periodic boundary conditions on the frequencies, with similar sizes and statistics of disordered samples leads to an estimate of νeff = 2β + γ = 1.2(2), that is compatible with a mean-field theory according to the condition (34). “
Page 12, second column, last paragraph: we modified “We find clear evidence for the onset of Replica Symmetry Breaking phase at the laser threshold studying the overlap distributions. “ in “We find clear evidence for the occurrence of a Replica Symmetry Breaking phase at low temperature. Studying the deviation of the overlap distributions from a Gaus- sian distribution by standard methods (e.g., the Binder cumulant), as performed in Ref. [40], the onset of such a spin-glass phase can be shown to occur at a tempera- ture consistent with the laser threshold identified by FSS analysis of the specific heat peaks. “
Page 14, appendix A, we added the last sentence “The statistical error on the average over disorder is much larger than the error on the thermal average and leads to the error bars on the observables displayed in the main text. “
New appendix B added.
Previous appendix B moved to appendix C.
Last equation of appendix C: absolute value of parameter C replaces C.

---

## Editorial Decision

resubmitted